# Fourier Features Let Agents Learn High Precision Policies with Imitation Learning

**Balázs Gyenes** [1,2]  **Emiliyan Gospodinov** [1]  **Jan Frieling** [1]  **Enrico Krohmer** [1]  **Nicolas Schreiber** [1]  **Xiaogang Jia** [1]
**Niklas Freymuth** [1]  **Gerhard Neumann** [1]

## Abstract

High-precision robotic manipulation requires fine-grained spatial reasoning that is often difficult to achieve with RGB-only policies due to depth ambiguity and perspective scale issues. Policies that leverage 3D information directly, such as those based on point clouds, offer a stronger geometric prior over purely image-based ones, yet their performance remains highly task-dependent. We hypothesize that this discrepancy may be due to the spectral bias of neural networks towards learning low frequency functions, which especially affects architectures conditioned on slow-moving Cartesian features. We thus propose to map point clouds from Cartesian space into high-dimensional Fourier space, effectively equipping the point cloud encoder with direct access to high-frequency features. We experimentally validate the use of Fourier features on challenging manipulation tasks from the RoboCasa and ManiSkill3 benchmarks and on a real robot setup. Despite their simplicity, we find that Fourier features provide significant benefits across diverse encoder architectures and benchmarks and are robust across hyperparameters. Our results indicate that Fourier features let policies leverage geometric details more effectively than Cartesian features, showing their potential as a general-purpose tool for point cloud-based imitation learning. We provide source code and videos on our project page: https://fourier-il.github.io/fourier-il.

[1]Autonomous Learning Robots, Karlsruhe Institute of Technology, Germany [2]HIDSS4Health - Helmholtz Information and Data Science School for Health, Karlsruhe/Heidelberg, Germany. Correspondence to: Balázs Gyenes <balazs@gyenes.ca>, Gerhard Neumann <gerhard.neumann@kit.edu>.

*Proceedings of the 43rd International Conference on Machine Learning*, Seoul, South Korea. PMLR 306, 2026. Copyright 2026 by the author(s).

## 1. Introduction

Diffusion-based Imitation Learning (IL) has emerged as a powerful framework for robotic visuomotor control (Chi et al., 2023; Reuss et al., 2023; Wu et al., 2025; Intelligence et al., 2025). By treating action generation as a denoising process (Ho et al., 2020), diffusion policies naturally capture the multi-modal action distributions of human expert demonstrations. This capability has made diffusion policies the state-of-the-art on long-horizon and multi-task manipulation benchmarks. Yet, the success of diffusion policies depends on the observation encoder's ability to extract subtle positional cues from the observed scene that inform the policy's next action. Policies that cannot respond to the geometric information hidden in observations are unable to imitate expert demonstrations that condition on this information.

RGB images remain the most common type of observation due to their semantic richness and the widespread availability of pretrained vision encoders. However, because they lack an explicit representation of 3D geometry, they require the policy to implicitly infer a 2D-to-3D mapping and are also sensitive to viewpoint and image artifacts, such as lighting variations (Ke et al., 2025; Wilcox et al., 2025; Donat et al., 2025). In contrast, 3D modalities such as depth maps, point clouds, and point maps directly represent shape, distance, and spatial relationships, helping the policy reason about geometry and occlusions and execute complex motions accurately. While this explicit structuring provides a strong geometric prior, the success of policies that rely purely on 3D modalities varies greatly between tasks. As a result, a number of hybrid 2D/3D architectures have been proposed (Ke et al., 2025; Wilcox et al., 2025; Goyal et al., 2023), which often use foundation models to extract features from the 2D RGB stream and combine this with 3D information in a variety of ways.

In this paper, we argue that the key shortcoming of current 3D modalities lies in the *spectral bias* of the architectures that are used to process them. For high-precision tasks, such as inserting a peg into a socket, the policy needs to learn a sharp decision boundary for e.g. whether to insert the peg or reposition it. Since the observations that should result in these respective actions differ only slightly, a high

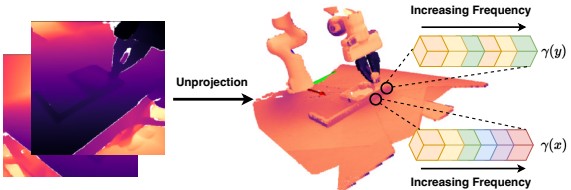

*Figure 1.* **Method Overview.** Adding a Fourier feature mapping from Cartesian coordinates into a higher-dimensional feature space improves performance for any point cloud encoder used for diffusion imitation learning. For high-precision policies, the network must learn to condition on fine details in the scene geometry e.g. to decide whether to insert the leg into the slot or reposition it, yet neural networks learn the high frequency components of the target function only slowly, if at all. While neighbouring points in the scene have very similar Cartesian features, the high-dimensional Fourier features allow them to easily be distinguished.

frequency function is best suited to represent this desired policy faithfully. Although neural networks are universal function approximators (Hornik et al., 1989), Multi-Layer Perceptrons (MLPs) and fully-connected layers have a spectral bias towards learning low-frequency components first, while high-frequency components converge slowly or may not be learned at all (Rahaman et al., 2019; Tancik et al., 2020). Although this phenomenon is well-known, these MLPs underpin the majority of point cloud architectures, which use them to encode Cartesian coordinates into latent features. In contrast, the convolutional layers underpinning most image-based architectures are inherently more biased towards high-frequency signals, with evidence that they may be lacking sensitivity to lower frequencies (Abello et al., 2021; Jo & Bengio, 2017; Wang et al., 2020).

In fields such as novel view synthesis, the spectral bias of MLPs is remedied using a Fourier feature mapping of the input coordinates (Mildenhall et al., 2021; Tancik et al., 2020). However, recent foundation models for robotic control from point clouds (Chen et al., 2024; Zhou et al., 2024; Jia et al., 2025b; Li et al., 2026) do not leverage this insight, and its use within IL architectures has so far been limited to isolated cases (Wilcox et al., 2025). This work therefore systematically evaluates Fourier feature mappings and their use for diffusion-based IL on point clouds across a broad range of policy architectures. By projecting points into a higher dimensional space where subtle geometric differences are amplified, we counteract spectral bias. Experimentally, we show that using Fourier-encoded input representations leads to consistent improvements across different point cloud architectures and benchmarks. These input representations improve success rate by up to $20\%$ and $7\%$ on RoboCasa (Nasiriany et al., 2024) and ManiSkill3 (Tao et al., 2025), respectively. Further, they improve normalized score on $4$ challenging real world tasks from $14.8\%$ to $40.2\%$. Qualitatively, policies trained with Fourier mappings exhibit smoother and more precise motions, particularly on robotic

control tasks where fine-grained manipulation matters.

Our contributions are as follows: **1)** we incorporating Fourier feature mappings into various point cloud encoders and show how this resolves their spectral bias; **2)** through experiments on a real robot and on the RoboCasa and ManiSkill task suites, we demonstrate consistent and robust improvements over baselines without Fourier feature mappings; and **3)** through extensive analysis and parameter studies, we provide a number of insights. Fourier features are most beneficial when point clouds contain rich geometric detail, but provide a boost even in the absence of fine geometry, perhaps by improving learning dynamics. Furthermore, Fourier features do not require additional regularization and are robust to choice of hyperparameters.

## 2. Related Work

**Imitation Learning in Robotics.** Recent progress in IL has been driven by incorporating diffusion (Chi et al., 2023; Reuss et al., 2023) and flow matching (Lipman et al., 2023) objectives, which enable policies to learn multi-modal action distributions, and by training policies on large-scale datasets (Black et al., 2024; Intelligence et al., 2025; Brohan et al., 2022; Zitkovich et al., 2023; Zhu et al., 2025) to significantly increase generalization and performance. However, these approaches are primarily conditioned on RGB images. This choice allows leveraging powerful pretrained visual encoders and provides strong semantic features, but lacks explicit representation of 3D geometry, causing issues with depth and scale ambiguity. Similarly, image-based representations are sensitive to viewpoint and lighting variations (Ze et al., 2024; Zhu et al., 2024; Wilcox et al., 2025).

**3D Visual Representations for Imitation Learning.** To address these shortcomings, 3D information can be leveraged in stand-alone modalities, such as point clouds or point maps, or in combination with RGB. On a number of challenging tasks, lightweight point cloud-based policies outperform RGB and RGB-D modalities while requiring significantly less data (Ze et al., 2024; Zhu et al., 2024; Ze et al., 2025). Point maps (Wang et al., 2024), an alternative to point clouds defined on a grid-like structure, have been proposed for IL (Jia et al., 2025a), but their performance can be inconsistent (Zhu et al., 2024). In contrast, a variety of hybrid 2D/3D approaches augment 2D features from pre-trained image encoders with 3D position information reconstructed from the original depth maps, either at the patch level (Ke et al., 2025), the point level (Wilcox et al., 2025), or in voxels (Gervet et al., 2023). This combines the benefits of large-scale image datasets with those of explicit geometric representations, emphasizing the benefits of complex architectural design. We argue that the recent popularity of such approaches is ultimately due to the spectral frequency bias of most point cloud encoders, which prevents

them from accurately learning high-precision tasks when employed in isolation. We show that simple architectures become significantly more effective when equipped with non-parametric Fourier input mappings.

**Deep Learning on Point Clouds** Since the advent of Point-Net (Qi et al., 2017a), many architectures have been proposed for processing unordered sets of points (Qi et al., 2017b; Zhao et al., 2021; Lai et al., 2022; Qian et al., 2022). A common paradigm inspired by vision transformers is to group the point clouds into patches which are then tokenized using a lightweight PointNet-style network (Pang et al., 2022; Yu et al., 2022; Chen et al., 2023). Patches are computed by taking the K-Nearest Neighors (KNN) around each patch center point, which are sampled randomly using Farthest Point Sampling (FPS). To reduce the dimensionality of the embedding, the patch tokens can optionally be aggregated using max pooling (Zhu et al., 2024) or attention (Gyenes et al., 2024). The flexibility of this paradigm has allowed it to be used to train multi-modal foundation models on point clouds (Chen et al., 2024; Zhou et al., 2024; Jia et al., 2025b; Li et al., 2026), but these models all operate on slow-changing Cartesian features. Because of the ubiquity of PointPatch architectures in the literature, we focus our experiments on several of its variants.

**Deep Learning with Fourier Features.** Neural networks have a spectral bias, i.e., a tendency to learn low-frequency components faster than high-frequency ones (Rahaman et al., 2019). As confirmed in more recent empirical work (Lippe et al., 2023; Würth et al., 2026), this is partly a characteristic of the mean-squared error (MSE) loss, which tends to focus on low-frequency signals in the data. Fourier features (Mildenhall et al., 2021; Tancik et al., 2020) mitigate this issue by projecting low-dimensional Cartesian coordinates into high-dimensional sinusoidal embeddings with multiple frequencies. This technique was key for enabling Neural Radiance Fields (NeRFs) to learn detailed 3D scenes with high fidelity and avoid blurry, oversmoothed reconstructions (Mildenhall et al., 2021; Tancik et al., 2020; Barron et al., 2022). The frequencies can be fixed or learned end-to-end (Gao et al., 2023; Sun et al., 2024). Adapt3R (Wilcox et al., 2025) leverages Fourier features to outperform other architectures on unseen viewpoints during inference, but they do not investigate their effect in other contexts. In contrast, we apply Fourier mappings systematically across point cloud architectures in diffusion-based IL, comparing them from a frequency-domain perspective.

## 3. Method

### 3.1. Problem Formulation

IL aims to learn a policy from expert demonstrations. We are given a dataset of $N$ expert trajectories

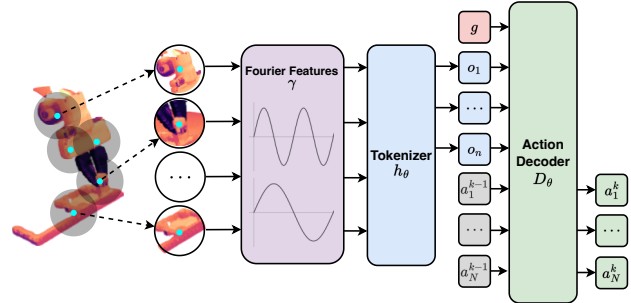

*Figure 2.* **Overview of PointPatch encoder family.** We group the input point cloud into neighborhoods $\mathcal{N}(i)$ (patch centers indicated in blue on the left). We map point coordinates into Fourier feature space to amplify subtle geometric differences between similar observations. The tokenizer extracts and aggregates features for each neighborhood to produce a set of tokens which are then forwarded to a goal-conditioned diffusion policy $D_\theta$ to denoise the next action chunk.

$\tau_i = (\mathbf{g}_i, (o_1, a_1), (o_2, a_2), \ldots, (o_{\ell_i}, a_{\ell_i}))$, where $\ell_i$ is the trajectory length and $\mathbf{g}_i$ is the language description for the trajectory. We aim to learn a policy $\pi(\mathbf{a}|o, \mathbf{g})$ that maps observations $o$ and embedded goal $\mathbf{g}$ to actions $\mathbf{a}$. Predicting sequences of actions, i.e. action chunking, where $\mathbf{a} = (a_k, a_{k+1}, \ldots, a_{k+H})$ with current time step $k$ and horizon $H$, results in more temporally correlated trajectories than predicting individual actions (Zhao et al., 2023). Each observation $o$ contains depth images from $M$ cameras. In combination with the camera intrinsic and extrinsic parameters from calibration, we can construct any desired 3D observation representation from these depth images.

### 3.2. Score-Based Diffusion

To learn policies from expert demonstrations, we use the typical Elucidated Diffusion Models (EDM) framework (Karras et al., 2022; Reuss et al., 2023) for score-based action diffusion conditioned on scene observations. Diffusion models learn to generate new samples by iteratively reversing a Gaussian perturbation process. Under this framework, the policy $\pi_\theta(\mathbf{a}|o)$ successively denoises actions generated from Gaussian noise back to the data manifold. The noising and its inverse process can be expressed with the following Stochastic Differential Equation (SDE)

$$d\mathbf{a}_\pm = \left(\pm\beta_t\sigma_t - \dot{\sigma}_t\right)\sigma_t\nabla_a \log p_t(\mathbf{a}|o, \mathbf{g})dt + \sqrt{2\beta_t}\sigma_t d\omega_t, \tag{1}$$

where $d\mathbf{a}_+$ and $d\mathbf{a}_-$ describe the forward and reverse process, respectively, $\beta_t$ determines the noise injection rate at diffusion time step $t$, $d\omega_t$ represents infinitesimal Gaussian noise, and $\nabla_a \log p_t(\mathbf{a}|o, \mathbf{g})$ denotes the score function of the diffusion process. To learn that score function, we train a neural network $D_\theta(\mathbf{a} + \boldsymbol{\epsilon}, o, \mathbf{g}, \sigma_t)$ via score matching (Vincent, 2011):

$$\mathcal{L}_{\text{SM}} = \mathbb{E}_{\sigma,\mathbf{a},\boldsymbol{\epsilon}}\big[\alpha(\sigma_t)|D_\theta(\mathbf{a} + \boldsymbol{\epsilon}, o, \mathbf{g}, \sigma_t) - \mathbf{a}|_2^2\big]. \tag{2}$$

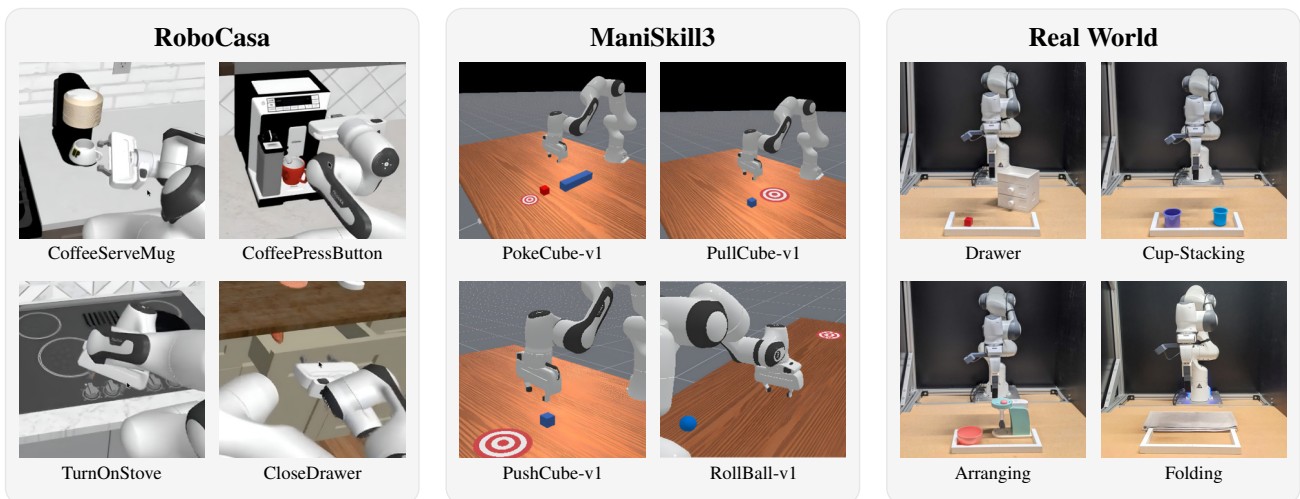

*Figure 3.* Overview of all evalution tasks from RoboCasa, ManiSkill3, and Real World benchmarks. **Left**: 4 of 16 RoboCasa tasks used for evaluation. **Middle**: all evaluated ManiSkill3 tasks. **Right**: starting configurations for all real-world tasks.

During policy sampling, i.e. during the reverse process, action samples are guided towards high-density regions of the data distribution by following the score function $\nabla_a \log p_t(\mathbf{a}|o, \mathbf{g})$. Therefore we can generate new action sequences beginning with Gaussian noise by iteratively denoising the action sequence with a numerical probability flow Ordinary Differential Equation (ODE) solver. We utilize the DDIM-solver to enable efficient action denoising in few steps (Song et al., 2021).

### 3.3. Point Clouds

Given a set of depth images from $M$ cameras $D^{(0)}, \ldots, D^{(M)} \in \mathbb{R}^{W \times H}$ as well as the camera intrinsics $K^{(0)}, \ldots, K^{(M)} \in \mathbb{R}^{3 \times 3}$, we first construct point clouds $X^{(j)} \in \mathbb{R}^{WH \times 3}$ in each camera's local coordinate frame via unprojection, i.e.,

$$X_{iW+j}^{(m)} = D_{i,j}^{(m)} \cdot (K^{(m)})^{-1} (i, j, 1)^{\mathbf{T}}. \qquad (3)$$

By multiplying each point cloud with its corresponding extrinsic matrix, we can transform it from the camera coordinate frame to the world frame. The final point cloud $X$ is obtained by concatenating point clouds from all $M$ views.

We treat point clouds as graphs, where the coordinates XYZ are the node features $\mathbf{x}^0$. This allows us to formulate the point cloud encoder as a message-passing Graph Neural Network (GNN) (Scarselli et al., 2009), a flexible framework that encompasses numerous well-known architectures. The point cloud encoder returns a tokenized embedding of the observed point cloud $\{\mathbf{T}_i\} \in \mathbb{R}^{n \times d}$. For PointPatch architectures, a positional encoding based on the patch center is also added to each token.

### 3.4. Fourier Feature Mapping

Neural networks are biased toward learning low-frequency components first, while high-frequency components converge slowly or may not be learned at all (Rahaman et al., 2019; Tancik et al., 2020). However, an IL policy parametrized by a neural network may need to learn a high frequency function to represent a sharp decision boundary, such as whether to reposition a grasped object or insert it. In 3D point clouds, a Fourier feature mapping allows the network to better distinguish points that are otherwise similar in Cartesian space. For a denoising diffusion model, this would allow the network to represent a score function that is a high-frequency function of the scene geometry, though not necessarily of the actions.

In contrast to previous work that adds Fourier features to specific, novel architectures (Wilcox et al., 2025), we hypothesize that applying a Fourier feature mapping to Cartesian points feature benefits essentially *any* point cloud-based policy. We adopt a NeRF-style, axis-aligned Fourier feature mapping (Mildenhall et al., 2021). Let $\mathbf{p} = (x, y, z) \in \mathbb{R}^3$ define a Cartesian point. The encoding function $\gamma : \mathbb{R} \to \mathbb{R}^{2L}$ applies sinusoids of different wavelengths $\lambda_k$ separately to the three coordinate values in $\mathbf{p}$ via the transformation

$$\gamma_k(x) = \left[\sin\left(\frac{2\pi x}{\lambda_k}\right), \cos\left(\frac{2\pi x}{\lambda_k}\right)\right]^{\mathbf{T}},$$
$$\lambda_k = \lambda_{\max} \left(\frac{\lambda_{\min}}{\lambda_{\max}}\right)^{\frac{k-1}{L-1}}, \qquad k = 1, \ldots, L, \qquad (4)$$

where $L$ is the number of frequency bands.

As the Fourier feature mapping is periodic, the point cloud must be bounded by the interval $[-\lambda_{\max}/2, \lambda_{\max}/2]$ to en-

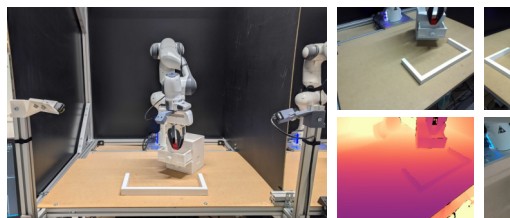

*Figure 4.* **Left**: Setup for the real-world drawer experiments. **Right**: RGB views from left, right and gripper cameras and depth view from the left camera.

sure unique features. If this is not possible, the input Cartesian coordinates can be concatenated with the Fourier features, which always yields a unique mapping.

### 3.5. Data Augmentation

The choice of wavelengths $\lambda_k$ is essential, as too short wavelengths may cause the network to overfit on the data, while too long wavelengths may not resolve the spectral bias (Tancik et al., 2020). Previous work using Fourier features for consistency models has even observed training instability with some hyperparameters (Song & Dhariwal, 2024). Instead of carefully tuning the wavelengths to each task, we choose a consistent set of wavelengths and use data augmentation to train the network to ignore frequencies that do not contain useful information. To achieve this, we apply VariableJitter (Gyenes et al., 2025), which samples the noise scale $\sigma \sim \mathcal{U}(0, \sigma_{\max})$ from a uniform distribution for each point cloud. Compared to uniform jitter, which applies noise $\epsilon \sim \mathcal{U}(-\sigma, \sigma)$ drawn from a fixed distribution to each point, this process avoids the difficulty of tuning the amplitude of typical uniform jitter. It ensures a trade-off between augmenting the data to reduce overfitting and ensuring there is no gap between training and testing data.

## 4. Experiments

### 4.1. Benchmarks and Datasets

We evaluate our approach on two widely used simulation benchmarks, RoboCasa (Nasiriany et al., 2024) and ManiSkill3 (Tao et al., 2025), as well as on four challenging real-world tasks (Jia et al., 2025a). Figure 3 visualizes exemplary tasks from all task suites. All models are trained in a multi-task setting, where the policy is provided a goal description in text form. RoboCasa and real world tasks include two static cameras and an in-hand camera, while ManiSkill3 tasks use only a static camera. In order to highlight the effect of Fourier features on 3D representations, we do not provide color features in simulation. Appendix A.1 provides additional details on all tasks.

**RoboCasa.** RoboCasa (Nasiriany et al., 2024) includes high-precision, long-horizon manipulation tasks in visually rich

kitchen scenes. We focus on 16 atomic tasks that stress fine geometric alignment and contact, which is where spectral bias is most detrimental. For each task we use 50 human-collected demonstrations provided by RoboCasa.

**ManiSkill3.** We further test on ManiSkill (Tao et al., 2025), where we evaluate on four tasks covering grasping and tool usage. Since the majority of tasks use color information to indicate some aspect of the target, we map the target's Cartesian coordinates to Fourier features and pass this as an additional goal token. We train on 500 demonstrations per task generated by an expert policies.

**Real World.** Finally, we adapt four challenging real-world tasks (Jia et al., 2025a) that feature long horizons, multiple phases, and precise manipulation. Figure 4 displays our setup, which consists of two static Zed Mini cameras and a RealSense D405 as an in-hand camera. Each task is comprised of distinct sub-tasks with different goal descriptions, such as "stack the red cup in the blue cup" for the "Cup-Stacking" task. For these complex tasks involving color information, we adopt a multi-modal approach utilizing RGB images and point clouds. Early experiments with unimodal methods were not successful. We use between 75 and 102 human demonstrations, depending on the task.

**Preprocessing.** Points beyond a maximum depth (2m in ManiSkill3 and the real world and 10m in RoboCasa) are removed. In ManiSkill and the real world, point clouds are further cropped to remove the irrelevant background points, and in ManiSkill, the table surface is also removed. We apply random downsampling to 32768 points followed by voxel downsampling with a voxel size of 1 cm. We sample a $\sigma_{\max}$ for VariableJitter up to 5 mm for ManiSkill, 2 mm for RoboCasa, and 1 mm for the real world, which augments the training data manifold while preserving crucial geometric details preserved. Camera observations are resized to $128 \times 128$ for ManiSkill tasks and $224 \times 224$ for RoboCasa.

### 4.2. Architectures and Baselines

We instantiate the denoising diffusion model as a decoder-only transformer. The transformer receives as input a goal token, a noise level token, $H$ noisy action tokens, where $H$ is the action horizon, and a number of observation tokens that are provided by a separate architecture, such as a point cloud encoder. A frozen CLIP RN-50 model (Radford et al., 2021) is used to embed the goal description into a token, while the noise level is encoded as in DDPM (Ho et al., 2020). A learned token position embedding is added onto each token, except for observation tokens when the number of tokens is variable. After the transformer layers, the final $H$ tokens are passed through a linear layer to arrive at denoised actions.

To highlight the effect of Fourier features for diffusion IL,

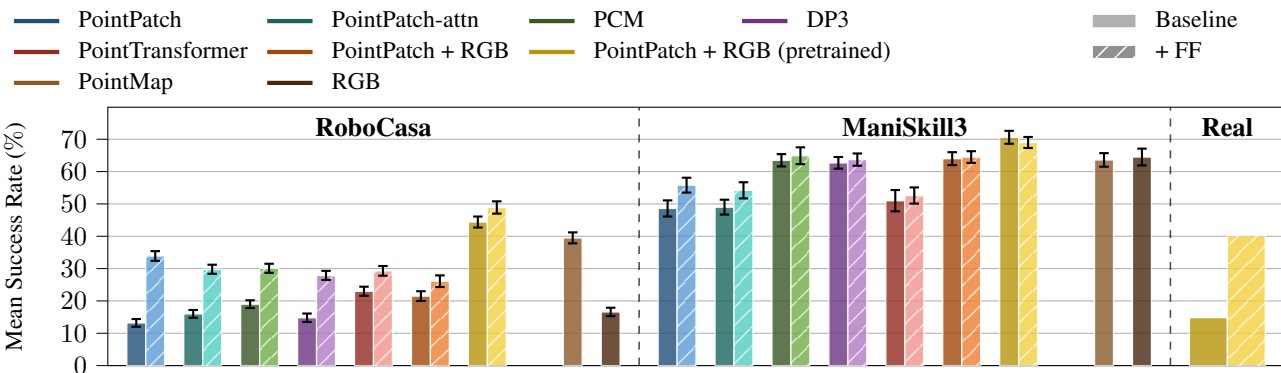

*Figure 5.* Mean success rate across all tasks of 3D encoders with and without Fourier features on RoboCasa (left), ManiSkill3 (middle), and the real world (right). Methods using Fourier Features are marked via hatched bars, and methods are displayed in the order of the legend at the top. Across diverse tasks and architectures, Fourier features provide a consistent and meaningful benefit to task performance.

we use the same diffusion backbone across all experiments and vary only the observation encoder across experiments. All architectures are instantiations of message neural networks with different aggregations and parameterizations for the learnable functions. We evaluate three variants as representative examples of the point-patching paradigm, which differ mainly in how they aggregate across patch tokens. **PointPatch** does not aggregate the patch tokens before the denoising step (Pang et al., 2022; Yu et al., 2022; Chen et al., 2023; Zhou et al., 2024; Chen et al., 2024), allowing each denoising step to attend to different parts of the observation. **PointPatch-attn** aggregates using attention pooling (Gyenes et al., 2024) to produce 3 tokens in all cases, which decreases computational cost. Lastly, the encoder referred to as "PointNet" in Point Cloud Matters (Zhu et al., 2024), which we call **PCM**, uses max pooling to aggregate across point patches. Furthermore, we apply Fourier features to existing architectures **DP3** (Ze et al., 2024), which applies max pooling across all points to create a single token, and **PointTransformer**, which uses attention to aggregate local regions iteratively. We make only minimal, necessary changes to each architecture, and apply Fourier feature mappings to absolute and/or relative point coordinates as required by the method. RGB images and pointmaps are tokenized using ConvNeXt V2 nano (Woo et al., 2023), a fully convolutional architecture. In **PointPatch + RGB**, we tokenize RGB images and point clouds separately, and concatenate tokens from the RGB modality to the tokens from the point cloud modality. More details on architectures are available in Appendix A.2.

### 4.3. Setup and Experiments.

**Research Questions.** Our experiments are designed to answer the following research questions: **Q1)** Do Fourier features yield consistent benefits across point cloud encoders and benchmarks? **Q2)** Does the benefit of Fourier feature

mappings translate from simulation to real-world tasks? **Q3)** How should these features be parameterized? **Q4)** How task-dependent are their effects?

To answer these questions, we first explore a series of simulation benchmarks and evaluate several point cloud encoders with and without Fourier features. Next, we train our policies on real-world data and evaluate them on a real robot. Finally, we then conduct an extensive parameter study and spectral analysis of our policies.

**Training and Evaluation.** Each method is trained with 5 random seeds, and we test performance at 3 checkpoints. We measure the average success rate across 50 rollouts for RoboCasa and 100 rollouts for ManiSkill and select the best-performing checkpoint for each seed. We use bootstrapping and report the interquartile mean with 95% confidence intervals. See Appendix A.4 for further details.

**Fourier Features.** We use a fixed set of $L$=16 frequency bands with log-spaced wavelengths between $\lambda_{max}$=4.0 m and $\lambda_{min}$=2.0 cm for all experiments. The choice of $\lambda_{max}$ ensures both the sin and cos components of the largest band are unique within the task space, which is typical bounded to roughly $[-2, 2]^3$, while $\lambda_{min}$ is small enough to discriminate neighboring voxels. The resulting $3 \times (2L)$=96 Fourier features per Cartesian point encode positions across this scale range, ranging from a global encoding at $\lambda_{max}$ to a voxel-level encoding at $\lambda_{min}$. See Appendix A.3 for additional hyperparameters.

## 5. Results

**Simulation Results.** Figure 5 shows average success rates over all tasks for RoboCasa and ManiSkill3. Tables 6 and 7 in the appendix provide detailed, per-task results for both benchmarks, respectively. In RoboCasa, we observe that Fourier feature mappings significantly boost both aggregate

success rate and success rates on a large number of individual tasks. For example, PointPatch on CloseDrawer improves from 34% to 72%, and TurnOffSinkFaucet improves from 28% to 63%, while the overall average increases from 13% to 34%. Similarly, the OpenDrawer task improves from almost no success at all to 12%. Despite using substantially different architectures, DP3 and PCM show similar trends, with both methods clearly benefitting from the use of Fourier features. While the relative ranking of different architectures is task-dependent, we find that the improvements from Fourier features are relatively consistent across architectures and task difficulties. We hypothesize that these improvements come from the Fourier mappings exposing high-frequency geometric cues to the denoising model, which alleviates their spectral bias and thus allows learning sharper, more meaningful token representations. Multimodal encoders that use RGB in conjunction with point clouds also benefit from Fourier features, despite the fact that the convolutional RGB tokenizer already has the capacity to represent high frequency functions. This highlights the potential for integrating Fourier features into large, multimodal models trained on internet-scale data. On ManiSkill3 tasks, we observe only minor improvements to PointPatch and PointPatch-attn but no significant improvement to other architectures. We hypothesize this is partly due to saturation on these relatively simple tasks.

**Real World Results.** We evaluate our best-performing method, PointPatch + RGB, with and without Fourier features for 16 rollouts on each real world task. Full results are in Appendix A.6. Table 8 shows that policies with Fourier features achieve significantly higher scores on all tasks, with an aggregate improvement from 14.8% to 40.23%. The policy without Fourier demonstrates especially poor performance on the Cup-Stacking task, often because it knocks the cup over while trying to grasp it. The least benefit is observed on the Folding task, which also requires the least geometric precision to carry out. These results demonstrate the Fourier features do not overfit on artifacts and are robust to real world conditions such as noisy depth measurements, occlusion artifacts, and camera miscalibration. Surprisingly, Fourier features greatly increase the performance of the PointPatch + RGB (pretrained) encoder on real world tasks, even though the performance gain on simulation tasks was more minor. This suggests that Fourier features remain effective even on more difficult tasks and when combined in more complex architectures. Table 9 shows further results on the Cup-Stacking task grouped by cup diameter. As cup size decreases, the benefit of Fourier features increases, albeit with the smallest cups, neither policy is able to solve the task consistently. This supports our claim that Fourier features allow the encoder to extract geometric details at smaller scales.

**Qualitative Results.** We find that policies trained on Fourier feature mappings move faster and more decisively, and more closely imitate the demonstration data. Policies trained without Fourier features tend to hesitate before making contact with objects, or behave as if they cannot perceive the scene. In the real world, the policies without Fourier features were often catastrophically unable to learn the data[1]. We hypothesize that due to the spectral bias of their MLP layers, these policies were overfitting on some global features of the observed point cloud rather than responding to the fine geometry near the end effector. See Appendix A.5 for a qualitative comparison of results on select RoboCasa tasks.

# 6. Analysis

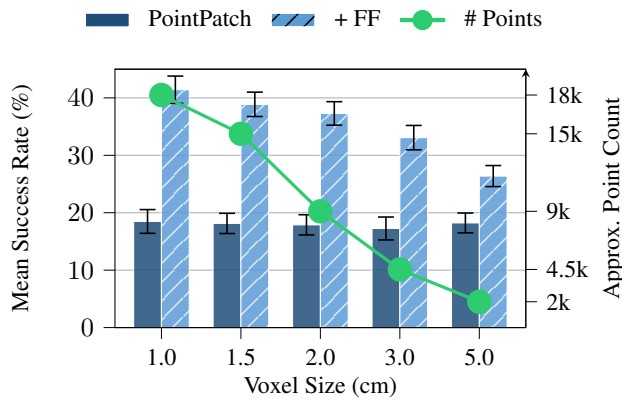

*Figure 6.* Success rates with and without Fourier features on point clouds of different sizes, achieved by voxel downsampling of the observations. Larger point clouds contain richer geometric detail, resulting in a larger benefit for Fourier features.

For additional experiments, we use a reduced set of 8 Robo-Casa tasks, utilizing the Pressing Buttons, Turning Levers, and Twisting Knobs task groups. Unless specified otherwise, we use the PointPatch encoder. We train a single policy on all 8 tasks, reporting the mean and 95% bootstrap confidence interval over 5 seeds for each method.

## 6.1. Frequency Analysis

**Point Cloud Size.** Since denser point clouds contain richer geometric detail, we expect Fourier features to provide a greater benefit with larger point clouds. To reduce point cloud sizes while retaining their overall structure, we increase the voxel size used for voxel downsampling. Figure 6 shows that point cloud size has a large effect on the performance of Fourier features, with the gap narrowing significantly for point clouds with 2k points. This demonstrates that Fourier features are more effective when point clouds contain more geometric detail to extract. Interestingly, the baseline method is essentially unaffected by the

---

[1]We provide rollout videos for real world tasks on our project page: https://fourier-il.github.io/fourier-il.

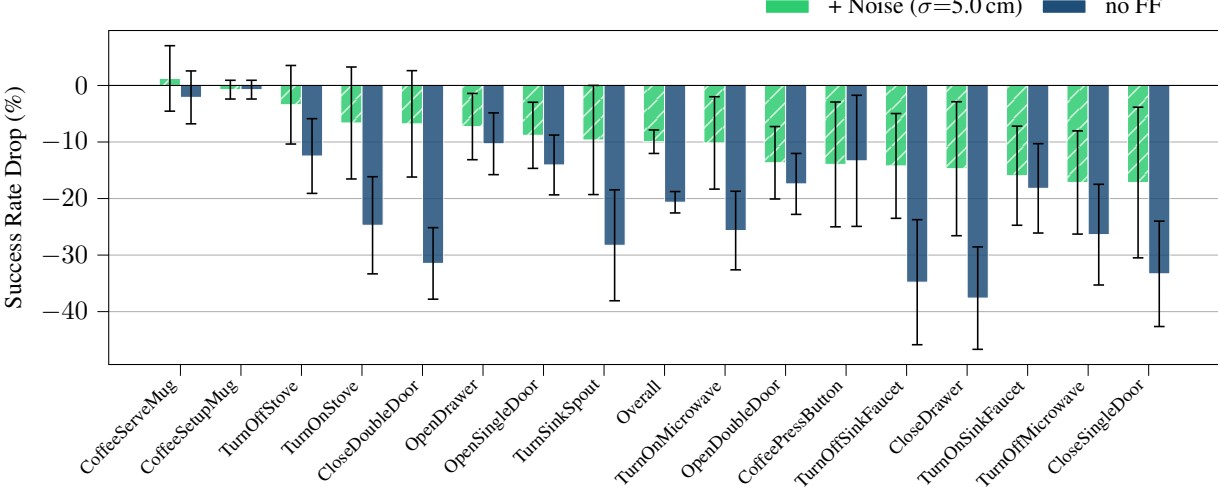

*Figure 7.* Absolute drop in success rate for each RoboCasa task resulting from removing Fourier features from the PointPatch policy architecture (**no FF**) or removing fine geometric information in the observation using Gaussian jitter (**+ Noise($\sigma$=5.0 cm)**). Even when high frequency information is removed, Fourier features still provide a meaningful benefit, perhaps by improving the learning dynamics of the policy.

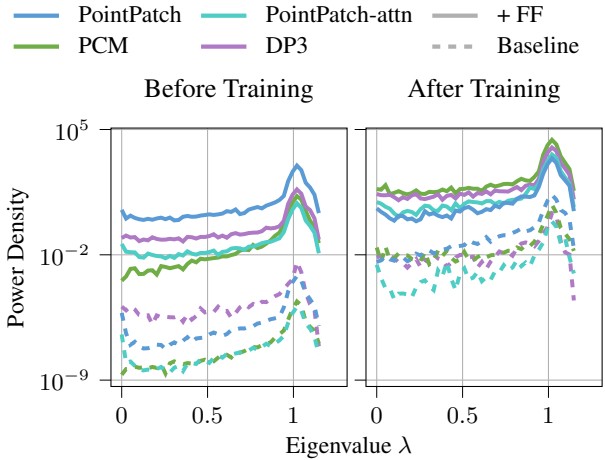

*Figure 8.* Graph Fourier spectra of the sensitivities of various architectures with respect to input point coordinates. During training, sensitivities increase by several orders of magnitude across all frequencies, and Fourier features also increase sensitivity by several more orders of magnitude relative to the baseline. The peak near eigenvalue of 1 indicates the orthogonal response, i.e. the isolated contribution of each point to the prediction.

heavy downsampling, suggesting that it does not condition on the geometric details that are removed by this process.

**Task-Dependent Fine Geometry.** The previous result raises the question - is there any observable pattern between the characteristics of a task and the effect of Fourier features? Is the advantage limited to tasks requiring fine manipulation? To investigate this, we train a policy with Fourier features and add Gaussian jitter with $\sigma$=5.0 cm to the point clouds during training and rollout. This extreme

level of jitter effectively removes any fine geometric information from the observation. We compare this against the drop in performance to that when Fourier features are not used.

Figure 7 shows that the advantage of Fourier features is not limited to tasks with fine geometry. There is only a weak correlation between the effect of removing the policy's ability to condition on geometric details (**no FF**) and removing the geometric details themselves from the observation (**+ Noise($\sigma$=5.0 cm)**). Furthermore, the policies with jitter still significantly outperform those without Fourier features, with an average success rate of 24% vs. 13% across Robo-Casa tasks. The broader benefit of Fourier features in the absence of fine geometry may be due to an improvement in learning dynamics in the policy.

**Spectral Density.** To show how architectural changes are reflected in the policy's sensitivity to spatial frequencies, we adapt a method from Miao et al. (2024). First, we create a "saliency point cloud", where each point contains the gradient of the predicted actions with respect to its coordinates. We take the Graph Fourier Transform (GFT) of this point cloud to quantify how rapidly this gradient varies between neighboring points, and visualize the spectrum of eigenvalues in Figure 8. We find that Fourier features increase the sensitivity of the network to high frequencies, as well as low ones. Since the sensitivites of all networks increase during training, Fourier features allow networks to learn the data more quickly. This suggests that the advantage of Fourier features may be due to their increased sensitivity to mid- and low-frequency bands as well. To further isolate this effect, Appendix A.7 also shows a spectral analysis of untrained PointNets on a synthetic unit sphere, confirming

that Fourier features shift the model's inductive bias toward high-frequency components ($\lambda > 1.0$) from initialization.

## 6.2. Parameter Studies

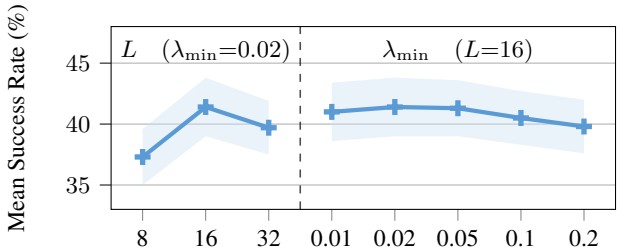

*Figure 9.* Parameter study of different Fourier feature wavelength configurations. Performance is robust to different numbers $L$ of log-spaced wavelengths (**left**), as well as to the minimum wavelength $\lambda_{\min}$ (**right**) around our default of $\lambda_{\min}=0.02$, $L=16$.

*Table 1.* Parameter study on jitter augmentation and Fourier feature encoding. Our method uses logspaced Fourier Features (FFs) and VariableJitter. **Top**: While VariableJitter slightls improves performance, data augmentation appears not to be essential. **Bottom**: Log-spaced, axis-aligned frequencies perform better than sampling randomly from a Gaussian (RFF). Learning the frequencies either directly or with Sinusoidal Positional Encoding (SPE) does not show a consistent benefit.

| Configuration | Mean Success Rate (%) |
|---|---|
| **Ours** | **$41.4 \pm 2.4$** |
| *Jitter Augmentation* | |
| no FFs, no jitter | $17.5 \pm 1.7$ |
| no FFs, random jitter | $17.0 \pm 1.6$ |
| no FFs, VariableJitter | $18.5 \pm 2.1$ |
| FFs, no jitter | $39.9 \pm 2.3$ |
| FFs, random jitter | $38.9 \pm 2.2$ |
| *Fourier Feature Encoding* | |
| log-spaced + SPE | $37.2 \pm 2.2$ |
| RFF | $24.0 \pm 2.0$ |
| RFF + learned | $22.9 \pm 1.8$ |
| RFF + SPE | $22.5 \pm 1.8$ |
| RFF + Cartesian | $23.4 \pm 1.9$ |

**Jitter.** To investigate the robustness of Fourier features, we consider different options for the jitter data augmentation, namely no jitter, standard random jitter drawn from $\sim \mathcal{U}(-\sigma_{\max}, \sigma_{\max})$, and VariableJitter as described in Subsection 3.5. Table 1 (**top**) shows that data augmentation may not be essential when using Fourier features, although VariableJitter may provide a minor boost over random jitter.

**Wavelengths.** We experiment with different numbers of wavelengths and a range of minimum wavelengths in Figure 9 (**middle**). While 16 wavelengths perform somewhat better than 8 or 32, Fourier features are robust to a wide range of minimum wavelengths. Unlike in related work

using Fourier features for NeRFs, we do not see any benefit to task-specific tuning of wavelength ranges (Sun et al., 2024). Together with the above, this consistency indicates that Fourier features used in diffusion IL are relatively insensitive to hyperparameters or changes in the training setup.

**Learned and Gaussian Fourier Features.** Sun et al. (2024) find that optimizing the frequencies directly through gradient descent is suboptimal, and they instead propose Sinusoidal Positional Encoding (SPE), which applies a linear layer and a sinusoidal non-linearity instead. Furthermore, instead of purely on-axis frequencies, Tancik et al. (2020) suggest randomly sampling frequency vectors $\mathbf{v} \sim \mathcal{N}(0, \sigma^2)$ from an isotropic normal distribution, where we set $\sigma=10$. In this case, the encoding function $\gamma : \mathbb{R}^3 \to \mathbb{R}^2$ applied to Cartesian point $\mathbf{p}$ is defined as $\gamma_k(\mathbf{p}) = [\sin(2\pi\mathbf{v}_k\cdot\mathbf{p}), \cos(2\pi\mathbf{v}_k\cdot\mathbf{p})]^{\mathbf{T}}$ for $k = 1, \ldots, L$. These Gaussian random Fourier features (RFFs) can likewise be learned either by directly optimizing frequencies or with SPE. Table 1 (**bottom**) evaluates different combinations of log-spaced and Gaussian RFFs with different methods for learning frequencies. We find that simply using a fixed encoding with log-spaced frequencies works best.

## 7. Conclusion

Neural networks are biased towards learning low-frequency functions of their inputs. In point cloud IL, this results in policies that ignore the high-frequency information that is essential for high-precision manipulation, such as insertion tasks or grasping. We incorporate the well-known Fourier feature mapping introduced in NeRF (Mildenhall et al., 2021) into a variety of point cloud-based Imitation Learning methods and test them on high-precision manipulation tasks in simulation and on a real robot.

We demonstrate that simply encoding the policy's coordinate inputs via Fourier features provides significant and consistent performance benefits. These benefits hold across RoboCasa and ManiSkill3 tasks of varying difficulty and are consistent across different point cloud encoders. Furthermore, performance also improves in multimodal architectures and are robust to real world noise and camera artifacts. Our analysis confirms that Fourier features are most helpful when point clouds contain rich geometric information, and seem to improve learning dynamics even in the absence of fine geometry. Parameter studies further show that Fourier features are robust to most hyperparameters, making them easy to use at essentially no additional cost. We thus argue that Fourier features should be used with practically any point cloud encoder architecture rather than Cartesian point features. Future work may investigate gradient-based learning of the optimal wavelengths or additional regularization to improve scalability.

## Acknowledgments

The present contribution is supported by the Helmholtz Association under the joint research school "HIDSS4Health – Helmholtz Information and Data Science School for Health. This work was supported by the European Research Council (ERC) under the European Union's Horizon Europe programme through the project SMARTI[3] (Grant Agreement No. 101171393). The authors gratefully acknowledge the computing time provided on the high-performance computer HoreKa by the National High-Performance Computing Center at KIT (NHR@KIT). This center is jointly supported by the Federal Ministry of Education and Research and the Ministry of Science, Research and the Arts of Baden-Württemberg, as part of the National High-Performance Computing (NHR) joint funding program. HoreKa is partly funded by the German Research Foundation (DFG). This work was supported by the Helmholtz Association's Initiative and Networking Fund on the HAICORE@KIT partition.

## Impact Statement

This paper presents work whose goal is to advance the field of machine learning. There are many potential societal consequences of our work, none of which we feel must be specifically highlighted here.

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

# A. Appendix

## A.1. Tasks

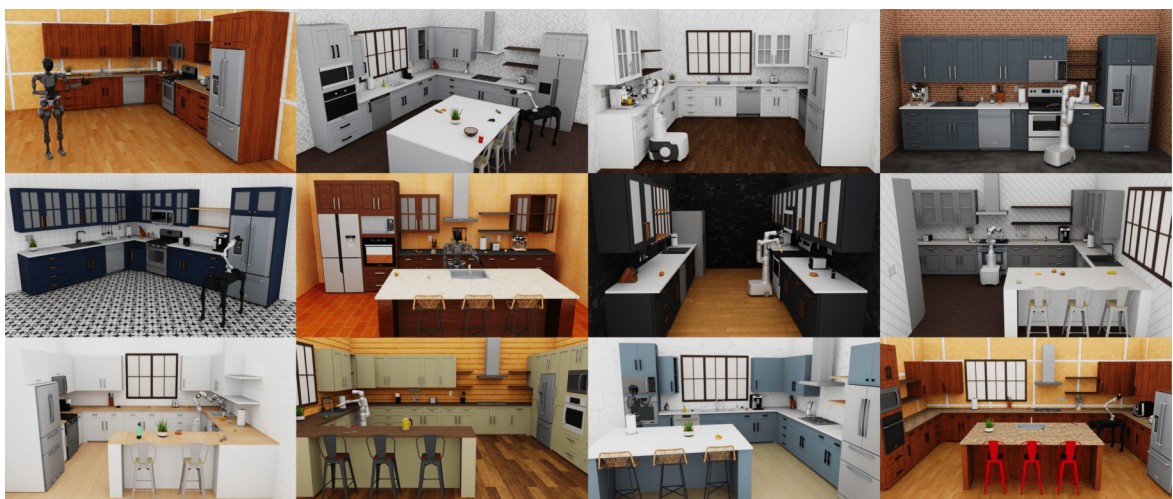

*Figure 10.* **Overview of RoboCasa Simulation Environments.** Example kitchen scenes and tasks illustrating the diversity of household manipulation settings provided by RoboCasa.

| Category | Task | Description |
|---|---|---|
| Insertion | CoffeeServeMug | Remove the mug from the holder and place it on the counter. |
| | CoffeeSetupMug | Place the mug into the coffee machine's mug holder. |
| Pressing Buttons | CoffeePressButton | Press the button to pour coffee into the mug. |
| | TurnOnMicrowave | Start the microwave by pressing the start button. |
| | TurnOffMicrowave | Stop the microwave by pressing the stop button. |
| Turning Levers | TurnOnSinkFaucet | Turn on the sink faucet to start water flow. |
| | TurnOffSinkFaucet | Turn off the sink faucet to stop water flow. |
| | TurnSinkSpout | Rotate the sink spout. |
| Twisting Knobs | TurnOnStove | Turn on a specific stove burner by twisting its knob. |
| | TurnOffStove | Turn off a specific stove burner by twisting its knob. |
| Open/Close Drawers | OpenDrawer | Open a drawer. |
| | CloseDrawer | Close a drawer. |
| Opening and Closing Doors | OpenSingleDoor | Open a microwave door or a cabinet with a single door. |
| | CloseSingleDoor | Close a microwave door or a cabinet with a single door. |
| | OpenDoubleDoor | Open a cabinet with two opposite-facing doors. |
| | CloseDoubleDoor | Close a cabinet with two opposite-facing doors. |

*Table 2.* RoboCasa evaluation tasks.

**RoboCasa.** RoboCasa (Nasiriany et al., 2024) is a large-scale simulation benchmark designed for training generalist robots in realistic household settings, with an emphasis on kitchen environments. It provides 100 tasks in total: 25 atomic tasks with 50 human demonstrations each, and 75 composite tasks with automatically generated demonstrations. The task set covers eight fundamental skills that are essential for home robotics: (1) pick-and-place, (2) door opening and closing, (3) drawer opening and closing, (4) knob turning, (5) lever manipulation, (6) button pressing, (7) insertion, and (8) navigation. The joint action space is 7-dimensional, including end-effector translation, rotation, and gripper control. For our experiments, we exclude the 8 Pick and Place tasks because they are too difficult, and the NavigateKitchen task because it has an incompatible action space, making multi-task training impractical. Table 2 lists each task along with the goal text used for it. Note that TurnSinkSpout, TurnOnStove, and TurnOffStove have dynamic goals that vary on each episode.

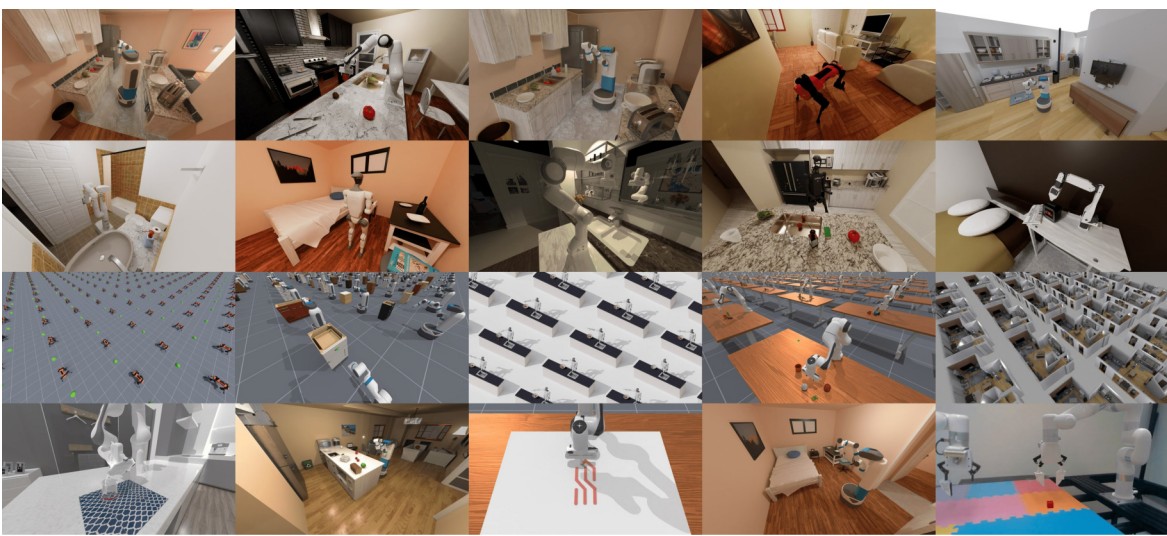

*Figure 11.* **Overview of ManiSkill3 Simulation Environments.** Example object-centric manipulation tasks illustrating the diversity of interactions supported by ManiSkill3.

*Table 3.* ManiSkill3 evaluation tasks.

| Category | Task | Description |
|---|---|---|
| Table-Top 2 Finger Gripper | PullCube-v1 | Pick up the cube and pull it to the target. |
| | PushCube-v1 | Push the cube into the target. |
| | PokeCube-v1 | Use the tool to poke the cube until it reaches the target. |
| | RollBall-v1 | Push the ball to make it roll into the target. |

**ManiSkill3.** ManiSkill3 (Tao et al., 2025) is a large-scale GPU-parallelized simulation benchmark designed for scalable training of embodied agents. It offers diverse object-centric manipulation tasks such as grasping, assembling, and tool use, with support for both imitation and reinforcement learning. Unlike RoboCasa, which emphasizes long-horizon household tasks in visually rich kitchen environments, ManiSkill3 provides highly parallelized simulation and rendering of physics-based interactions, enabling efficient large-scale experimentation and evaluation of manipulation policies. A summary of all ManiSkill3 tasks can be found in Table 3, each representing a distinct skill.

*Table 4.* Real world tasks with their goal descriptions.

| Task | Descriptions |
|------|--------------|
| Drawer | put the red cube in the top drawer |
| | put the red cube in the bottom drawer |
| Cup-Stacking | stack the blue cup in the purple cup |
| | stack the orange cup in the red cup |
| | stack the red cup in the blue cup |
| | stack the yellow cup in the orange cup |
| Arranging | put the blue cup in the coffee machine |
| | put the orange cup in the coffee machine |
| | put the pink bowl in the coffee machine |
| | put the red cup in the coffee machine |

**Real World.** A summary of the real world tasks along with their various goal descriptions can be found in Table 4.

## A.2. Architectures

**PointPatch.** The PointPatch encoder (Pang et al., 2022; Yu et al., 2022; Gyenes et al., 2024) divides a given point cloud into overlapping patches, tokenizes each patch and then uses a transformer for token processing. Point features are the Cartesian coordinates relative to the patch center, and are encoded with a lightweight PointNet (Qi et al., 2017b) to create patch tokens. Token position embeddings are computed by passing each centroid position through a two-layer MLP (Pang et al., 2022). We apply Fourier feature projections to the relative patch coordinates as well as the centroid positions.

**PointCloudMatters Encoder.** We additionally evaluate the PointCloudMatters-PointNet (PCM) encoder used in Zhu et al. (2024). This architecture is based on the patching paradigm introduced in PointMAE (Pang et al., 2022), but applies a max aggregation across patches, followed by a final projection head to compute a single token for the entire point cloud. In addition, each point is assigned its absolute Cartesian coordinates as well as its relative coordinates within the patch as features.

**DP3 Encoder.** Unlike patch-based methods, the DP3 encoder (Ze et al., 2024) creates a single token that embeds information from the entire point cloud. Point features are passed through an MLP, followed by a max-pooling operation to obtain order-invariant global features. A final projection head maps the embedding to the token dimension, resulting in $\mathbf{T} \in \mathbb{R}^{1 \times D}$. Although this architecture is simple, it is quite data efficient due to its small number of parameters. Unlike in the authors' implementation, we do not apply FPS sampling before the encoder since this decreased performance in our experiments.

**PointTransformer Encoder.** Inspired by the success of attention-based in methods in vision models, PointTransformer (Zhao et al., 2021) applies attention in a hierarchical manner. PointTransformer blocks augment point features using self-attention between points within a local neighborhood. These local neighborhoods are computed with FPS and KNN. This blocks alternate with down-sampling blocks that reduce the number of points and increase feature size. We use the classification variety, which outputs a single token from the point cloud. Since the node position inputs are used for FPS and KNN queries, we do not modify this input in any way. Instead, we also pass the point positions as the node features, with a Fourier feature projection where applicable. Because of the high compute costs of this architecture, we train each policy on 2-3 tasks instead of the typical 8.

**Pointmap Encoder.** To compare against alternative 3D representations, we also evaluate point maps (Wang et al., 2024; Jia et al., 2025a), which contain the same information as point clouds but are arranged in a 2D grid. Given depth images from multiple cameras and their intrinsics and extrinsics parameters, we unproject each pixel into 3D and transform it into the world frame, resulting in a dense point map $\mathbf{X} \in \mathbb{R}^{H \times W \times 3}$ for each camera. The resulting 3D representation can be processed directly with convolutional backbones such as ConvNeXt V2 (Woo et al., 2023) or ResNet (He et al., 2015).

**RGB+PointPatch Encoder.** Due to the flexible transformer design of our denoising model, multimodal observations can be processed by concatenating tokens from parallel observation encoders. We encode the RGB stream from each camera into a token with a ConvNeXt V2 nano ($\sim$15M parameters) (Woo et al., 2023) that is initialized from a pretrained checkpoint. Point clouds are processed by the PointPatch encoder.

## A.3. Hyperparameters

*Table 5.* Summary of select hyperparameters for simulation experiments.

| Hyperparameter | ManiSkill | RoboCasa |
|---|---|---|
| Training Epochs | 150 | 50 |
| Number of Attention Blocks | 4 | 4 |
| Attention Heads | 4 | 4 |
| Action Chunk Size | 10 | 20 |
| History Length | 1 | 1 |
| Embedding Dimension | 256 | 256 |
| Goal Lang Encoder | CLIP Resnet-50 | CLIP Resnet-50 |
| Attention Dropout | 0.3 | 0.3 |
| Residual Dropout | 0.1 | 0.1 |
| MLP Dropout | 0.1 | 0.1 |
| Optimizer | AdamW | AdamW |
| Betas | [0.9, 0.9] | [0.9, 0.9] |
| Learning Rate | 1e-4 | 1e-4 |
| Weight Decay | 0.05 | 0.05 |
| $\sigma_{max}$ | 80 | 80 |
| $\sigma_{min}$ | 0.001 | 0.001 |
| $\sigma_t$ | 0.5 | 0.5 |
| EMA decay | 0.995 | 0.995 |
| Time steps | Exponential | Exponential |
| Sampler | DDIM | DDIM |
| Denoising Steps | 10 | 10 |

*Table 6.* Average success rates on different RoboCasa tasks across 5 seeds. Fourier features generally lead to significant improvements for both PointPatch, DP3, and PCM architectures. In contrast, the convolutional PointMap struggles on these tasks, likely due to task complexity and data sparsity.

| Category | Task | PointPatch | + FF | PointPatch-attn | + FF | PCM | + FF | DP3 | + FF |
|---|---|---|---|---|---|---|---|---|---|
| Insertion | CoffeeServeMug | $1.0 \pm 2.2$ | $3.1 \pm 4.1$ | $1.8 \pm 2.6$ | $3.5 \pm 2.5$ | $2.7 \pm 2.9$ | $5.2 \pm 4.0$ | $1.0 \pm 2.2$ | $1.5 \pm 2.5$ |
| | CoffeeSetupMug | $0.0 \pm 0.0$ | $0.7 \pm 1.7$ | $0.0 \pm 0.0$ | $0.2 \pm 1.4$ | $0.0 \pm 0.0$ | $0.0 \pm 0.0$ | $0.0 \pm 0.0$ | $0.0 \pm 0.0$ |
| Pressing Buttons | CoffeePressButton | $19.8 \pm 7.0$ | $33.1 \pm 9.3$ | $16.1 \pm 4.3$ | $24.2 \pm 5.4$ | $22.2 \pm 5.4$ | $23.4 \pm 5.4$ | $18.6 \pm 5.8$ | $20.6 \pm 5.4$ |
| | TurnOnMicrowave | $7.8 \pm 3.4$ | $\mathbf{33.5 \pm 6.1}$ | $9.4 \pm 5.0$ | $\mathbf{32.1 \pm 5.5}$ | $21.0 \pm 5.4$ | $27.6 \pm 5.2$ | $20.0 \pm 6.0$ | $\mathbf{33.9 \pm 7.3}$ |
| | TurnOffMicrowave | $16.6 \pm 5.0$ | $\mathbf{42.9 \pm 7.3}$ | $18.6 \pm 7.8$ | $38.0 \pm 5.6$ | $22.6 \pm 5.8$ | $39.0 \pm 5.8$ | $21.1 \pm 6.1$ | $42.7 \pm 8.9$ |
| Turning Levers | TurnOnSinkFaucet | $16.2 \pm 4.6$ | $34.4 \pm 6.4$ | $13.9 \pm 4.1$ | $25.7 \pm 6.7$ | $20.8 \pm 5.2$ | $32.5 \pm 7.5$ | $18.8 \pm 5.2$ | $34.2 \pm 5.8$ |
| | TurnOffSinkFaucet | $28.0 \pm 8.4$ | $\mathbf{62.8 \pm 7.2}$ | $35.2 \pm 6.0$ | $54.4 \pm 5.2$ | $52.2 \pm 7.4$ | $54.4 \pm 6.0$ | $37.2 \pm 9.6$ | $61.0 \pm 6.2$ |
| | TurnSinkSpout | $36.3 \pm 6.9$ | $\mathbf{64.6 \pm 7.0}$ | $41.2 \pm 5.6$ | $50.0 \pm 5.6$ | $56.4 \pm 5.6$ | $60.7 \pm 6.3$ | $45.8 \pm 7.4$ | $54.6 \pm 6.6$ |
| Twisting Knobs | TurnOnStove | $14.6 \pm 5.4$ | $\mathbf{39.3 \pm 6.7}$ | $24.1 \pm 6.3$ | $35.4 \pm 5.8$ | $29.8 \pm 5.0$ | $38.6 \pm 6.6$ | $21.4 \pm 5.8$ | $35.4 \pm 5.8$ |
| | TurnOffStove | $7.9 \pm 4.1$ | $\mathbf{20.4 \pm 5.2}$ | $12.2 \pm 4.6$ | $19.7 \pm 5.9$ | $15.1 \pm 4.9$ | $17.6 \pm 6.0$ | $9.8 \pm 4.2$ | $16.4 \pm 4.0$ |
| Open/Close Drawers | OpenDrawer | $1.3 \pm 1.9$ | $\mathbf{11.7 \pm 5.1}$ | $3.6 \pm 2.4$ | $8.4 \pm 4.8$ | $3.4 \pm 2.2$ | $11.1 \pm 5.3$ | $3.8 \pm 3.0$ | $3.3 \pm 3.1$ |
| | CloseDrawer | $33.9 \pm 6.5$ | $\mathbf{71.5 \pm 6.3}$ | $38.6 \pm 8.6$ | $62.1 \pm 8.3$ | $23.8 \pm 5.4$ | $57.2 \pm 7.2$ | $27.6 \pm 6.4$ | $53.2 \pm 6.0$ |
| Open/Close Doors | OpenSingleDoor | $0.3 \pm 2.1$ | $14.4 \pm 4.9$ | $3.2 \pm 3.2$ | $\mathbf{17.5 \pm 6.1}$ | $3.7 \pm 3.1$ | $11.9 \pm 5.3$ | $0.2 \pm 1.4$ | $11.6 \pm 4.8$ |
| | CloseSingleDoor | $24.8 \pm 5.6$ | $\mathbf{58.1 \pm 7.5}$ | $19.6 \pm 6.4$ | $57.0 \pm 7.8$ | $13.6 \pm 6.0$ | $56.7 \pm 5.9$ | $6.3 \pm 5.3$ | $40.5 \pm 6.5$ |
| | OpenDoubleDoor | $0.0 \pm 0.0$ | $\mathbf{17.4 \pm 5.4}$ | $0.0 \pm 0.0$ | $13.7 \pm 5.5$ | $0.7 \pm 1.7$ | $9.8 \pm 4.6$ | $0.0 \pm 0.0$ | $10.5 \pm 4.3$ |
| | CloseDoubleDoor | $1.5 \pm 2.5$ | $33.0 \pm 5.8$ | $18.0 \pm 5.2$ | $34.0 \pm 7.2$ | $15.6 \pm 6.0$ | $34.2 \pm 7.0$ | $3.4 \pm 3.0$ | $26.6 \pm 5.4$ |
| **Overall** | | $13.2 \pm 1.2$ | $\mathbf{33.9 \pm 1.5}$ | $16.0 \pm 1.2$ | $\mathbf{29.8 \pm 1.4}$ | $19.0 \pm 1.2$ | $\mathbf{30.1 \pm 1.4}$ | $14.8 \pm 1.3$ | $\mathbf{27.9 \pm 1.4}$ |

| Category | Task | PointTransformer | + FF | PointPatch+RGB | + FF | PointPatch+RGB (pretrained) | + FF | PointMap | RGB |
|---|---|---|---|---|---|---|---|---|---|
| Insertion | CoffeeServeMug | $1.4 \pm 2.2$ | $3.2 \pm 2.8$ | $13.6 \pm 5.2$ | $14.7 \pm 6.1$ | $43.7 \pm 7.5$ | $49.1 \pm 7.7$ | $38.8 \pm 8.4$ | $3.8 \pm 3.0$ |
| | CoffeeSetupMug | $0.0 \pm 0.0$ | $1.5 \pm 2.5$ | $0.2 \pm 1.4$ | $1.8 \pm 2.6$ | $10.2 \pm 3.8$ | $9.2 \pm 4.0$ | $5.5 \pm 2.9$ | $0.3 \pm 2.1$ |
| Pressing Buttons | CoffeePressButton | $23.4 \pm 5.4$ | $25.8 \pm 5.8$ | $40.8 \pm 8.4$ | $40.5 \pm 8.3$ | $81.1 \pm 7.3$ | $84.6 \pm 7.0$ | $43.2 \pm 8.4$ | $37.1 \pm 6.1$ |
| | TurnOnMicrowave | $18.8 \pm 7.2$ | $\mathbf{33.5 \pm 5.7}$ | $26.0 \pm 8.4$ | $33.6 \pm 10.4$ | $43.3 \pm 7.5$ | $52.2 \pm 11.0$ | $36.2 \pm 7.8$ | $20.6 \pm 5.8$ |
| | TurnOffMicrowave | $24.2 \pm 5.8$ | $\mathbf{37.8 \pm 6.6}$ | $20.4 \pm 5.2$ | $37.9 \pm 14.7$ | $51.0 \pm 10.2$ | $52.0 \pm 11.2$ | $46.5 \pm 6.9$ | $10.9 \pm 4.3$ |
| Turning Levers | TurnOnSinkFaucet | $26.4 \pm 5.6$ | $\mathbf{44.2 \pm 7.4}$ | $24.2 \pm 7.0$ | $26.5 \pm 6.7$ | $54.3 \pm 5.7$ | $60.0 \pm 7.2$ | $34.6 \pm 5.4$ | $16.5 \pm 6.3$ |
| | TurnOffSinkFaucet | $55.9 \pm 6.9$ | $59.0 \pm 9.4$ | $29.4 \pm 5.4$ | $37.2 \pm 7.2$ | $44.6 \pm 6.6$ | $55.9 \pm 8.1$ | $46.2 \pm 9.0$ | $22.1 \pm 5.1$ |
| | TurnSinkSpout | $51.6 \pm 7.2$ | $62.9 \pm 7.7$ | $29.3 \pm 9.5$ | $34.1 \pm 5.9$ | $5.0 \pm 3.0$ | $7.4 \pm 4.2$ | $43.1 \pm 8.5$ | $24.4 \pm 7.2$ |
| Twisting Knobs | TurnOnStove | $33.6 \pm 6.4$ | $37.4 \pm 5.4$ | $13.6 \pm 5.6$ | $13.6 \pm 4.8$ | $19.5 \pm 5.9$ | $22.4 \pm 5.2$ | $18.9 \pm 6.7$ | $10.1 \pm 3.5$ |
| | TurnOffStove | $16.6 \pm 4.6$ | $20.7 \pm 4.9$ | $6.6 \pm 3.0$ | $7.7 \pm 3.1$ | $10.3 \pm 4.9$ | $13.3 \pm 4.7$ | $11.7 \pm 5.5$ | $4.9 \pm 3.1$ |
| Open/Close Drawers | OpenDrawer | $5.8 \pm 3.4$ | $\mathbf{15.6 \pm 6.0}$ | $7.2 \pm 4.0$ | $14.6 \pm 4.6$ | $28.1 \pm 6.7$ | $26.8 \pm 6.8$ | $40.0 \pm 7.2$ | $6.5 \pm 3.1$ |
| | CloseDrawer | $57.6 \pm 8.8$ | $57.0 \pm 9.8$ | $49.8 \pm 7.4$ | $54.5 \pm 10.3$ | $76.6 \pm 5.0$ | $80.4 \pm 5.2$ | $95.1 \pm 2.7$ | $41.0 \pm 7.4$ |
| Open/Close Doors | OpenSingleDoor | $14.4 \pm 4.0$ | $13.3 \pm 5.5$ | $29.2 \pm 6.0$ | $27.4 \pm 8.6$ | $45.2 \pm 7.2$ | $49.8 \pm 5.8$ | $49.0 \pm 7.4$ | $24.7 \pm 5.9$ |
| | CloseSingleDoor | $30.2 \pm 8.2$ | $33.3 \pm 6.3$ | $42.4 \pm 6.8$ | $52.0 \pm 6.8$ | $74.2 \pm 5.4$ | $73.2 \pm 7.2$ | $67.7 \pm 5.5$ | $40.0 \pm 8.4$ |
| | OpenDoubleDoor | $1.9 \pm 2.9$ | $6.6 \pm 3.8$ | $1.3 \pm 1.9$ | $4.9 \pm 3.1$ | $61.9 \pm 10.3$ | $74.0 \pm 14.8$ | $21.8 \pm 7.0$ | $0.7 \pm 1.7$ |
| | CloseDoubleDoor | $5.7 \pm 4.3$ | $\mathbf{16.7 \pm 4.5}$ | $9.3 \pm 5.1$ | $15.9 \pm 5.3$ | $60.8 \pm 7.2$ | $72.5 \pm 9.7$ | $34.0 \pm 7.6$ | $1.2 \pm 2.0$ |
| **Overall** | | $23.0 \pm 1.4$ | $\mathbf{29.3 \pm 1.5}$ | $21.5 \pm 1.5$ | $\mathbf{26.1 \pm 1.8}$ | $44.4 \pm 1.7$ | $\mathbf{48.9 \pm 1.9}$ | $39.5 \pm 1.7$ | $16.6 \pm 1.3$ |

## A.4. Simulation Results

Tables 6 and 7 show per-task success rates for RoboCasa and Maniskill real tasks, respectively. As in Jia et al. (2025a), we test policies at checkpoints after 60%, 80%, and 100% of training epochs (30, 40, and 50 epochs for RoboCasa, and 90, 120, and 150 epochs for ManiSkill). We perform 50 rollouts on RoboCasa tasks and 100 rollouts on ManiSkill tasks, compute the average success rate and select the best-performing checkpoint for each seed. The final success rate is the mean over 5 random seeds.

To compute confidence bounds, we adapt the bootstrapping method from Agarwal et al. (2021) and report the interquartile mean (ICM) and 95% confidence bounds. To simulate one experiment, we sample 5 seeds with replacement. For each checkpoint of each seed, we sample episodes with replacement (50 for RoboCasa and 100 for ManiSkill), then select the best checkpoints and recompute the mean success. After repeating this for 50,000 simulated experiments, the interquartile mean is the average success after discarding the top and bottom quartiles, while the 95% confidence bounds are the values that 95% of experiments fall between.

*Table 7.* Average success rates on Maniskill tasks across 5 seeds. Fourier features improve the performance of point-cloud based architectures, likely because they enable better differentiation of fine-grained details. For Maniskill, PointMaps are competitive with approaches enhanced with Fourier features, presumably due to larger training datasets. **Bold** denotes statistical significance.

| Category | Task | PointPatch | + FF | PointPatch-attn | + FF | PCM | + FF | DP3 | + FF |
|---|---|---|---|---|---|---|---|---|---|
| Table-Top 2 Finger Gripper | PullCube-v1 | $51.4 \pm 4.6$ | $57.1 \pm 4.3$ | $46.8 \pm 6.0$ | $52.9 \pm 5.3$ | $77.3 \pm 3.9$ | $78.8 \pm 3.8$ | $80.0 \pm 4.4$ | $78.3 \pm 3.9$ |
| | PushCube-v1 | $69.3 \pm 5.3$ | $75.7 \pm 5.7$ | $69.3 \pm 4.5$ | $74.5 \pm 5.1$ | $78.4 \pm 3.8$ | $81.4 \pm 5.4$ | $79.3 \pm 3.1$ | $81.8 \pm 3.4$ |
| | PokeCube-v1 | $55.4 \pm 5.4$ | $63.6 \pm 4.2$ | $56.4 \pm 4.2$ | $63.5 \pm 5.9$ | $71.0 \pm 3.6$ | $69.2 \pm 6.6$ | $68.2 \pm 4.0$ | $66.9 \pm 3.7$ |
| | RollBall-v1 | $18.2 \pm 5.0$ | $26.8 \pm 4.2$ | $23.6 \pm 3.6$ | $25.8 \pm 4.0$ | $27.1 \pm 4.1$ | $30.0 \pm 4.6$ | $23.3 \pm 3.3$ | $27.7 \pm 4.3$ |
| **Overall** | | $48.6 \pm 2.5$ | $\mathbf{55.8 \pm 2.3}$ | $49.0 \pm 2.3$ | $\mathbf{54.2 \pm 2.5}$ | $63.5 \pm 1.9$ | $64.9 \pm 2.6$ | $62.7 \pm 1.8$ | $63.7 \pm 1.9$ |

| Category | Task | PointTransformer | + FF | PointPatch+RGB | + FF | PointPatch+RGB (pretrained) | + FF | PointMap | RGB |
|---|---|---|---|---|---|---|---|---|---|
| Table-Top 2 Finger Gripper | PullCube-v1 | $57.5 \pm 8.5$ | $63.9 \pm 5.3$ | $75.4 \pm 4.8$ | $74.5 \pm 3.9$ | $82.7 \pm 3.9$ | $80.6 \pm 3.6$ | $71.7 \pm 4.3$ | $74.3 \pm 6.9$ |
| | PushCube-v1 | $74.5 \pm 6.7$ | $73.9 \pm 4.1$ | $82.7 \pm 3.1$ | $81.2 \pm 3.0$ | $91.2 \pm 3.0$ | $88.6 \pm 2.6$ | $81.2 \pm 4.2$ | $79.4 \pm 4.0$ |
| | PokeCube-v1 | $57.8 \pm 5.0$ | $58.1 \pm 6.3$ | $69.9 \pm 3.7$ | $75.3 \pm 4.3$ | $76.5 \pm 3.9$ | $74.8 \pm 3.8$ | $69.8 \pm 4.4$ | $72.9 \pm 4.9$ |
| | RollBall-v1 | $14.1 \pm 6.9$ | $14.4 \pm 4.4$ | $27.9 \pm 4.3$ | $27.0 \pm 3.8$ | $32.0 \pm 5.4$ | $31.7 \pm 4.3$ | $31.5 \pm 4.1$ | $31.3 \pm 5.3$ |
| **Overall** | | $51.0 \pm 3.3$ | $52.6 \pm 2.5$ | $64.0 \pm 2.0$ | $64.5 \pm 1.8$ | $70.6 \pm 2.0$ | $69.0 \pm 1.7$ | $63.6 \pm 2.1$ | $64.5 \pm 2.6$ |

## A.5. Qualitative Results

Figure 12 shows representative rollouts from PointPatch+FF policies on selected RoboCasa tasks. Overall, the agents trained with Fourier features reliably make contact with the target objects (e.g., buttons and lever handles) and completes all three tasks, whereas the agents trained without Fourier features fail to accomplish the tasks.

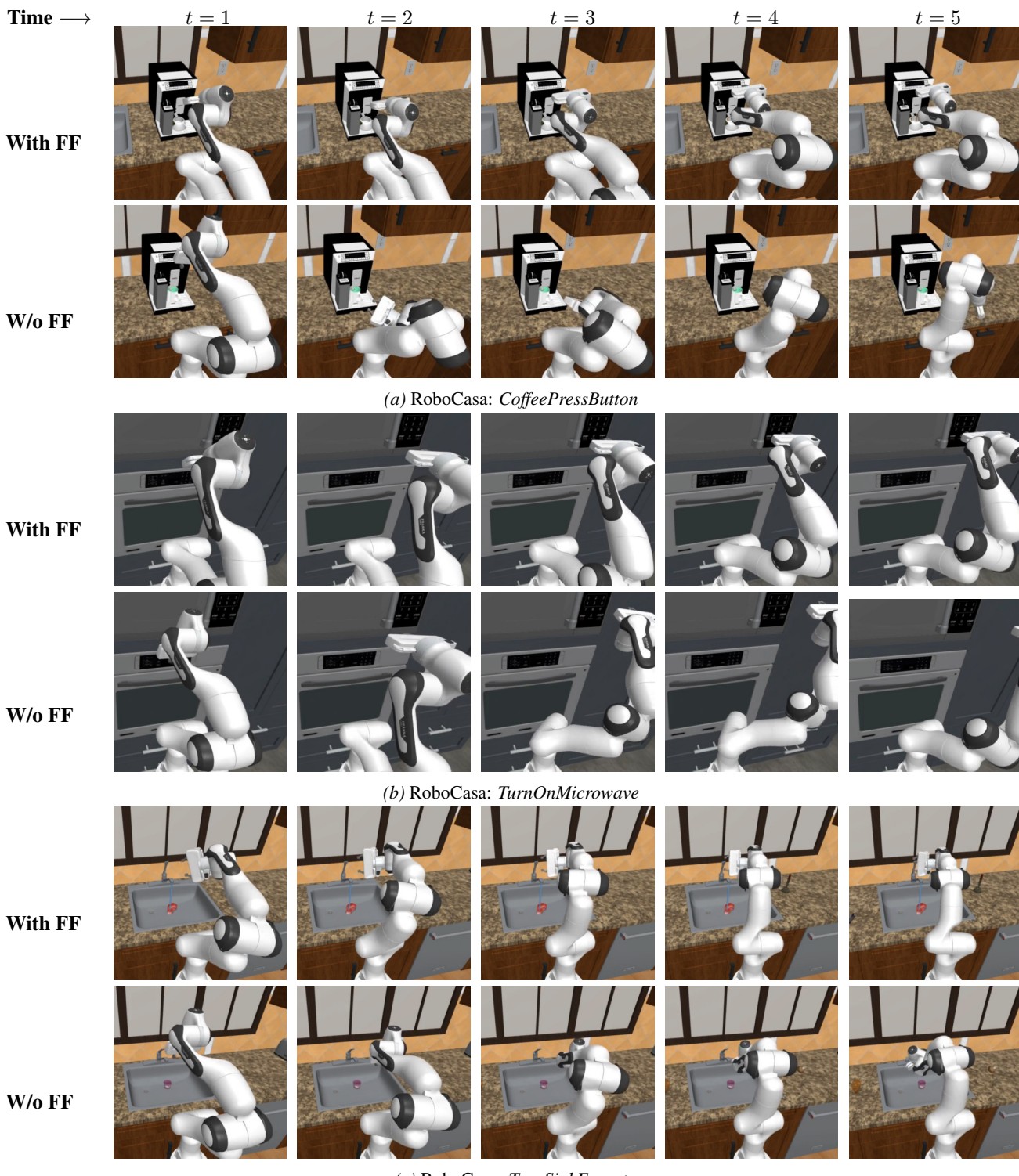

*Figure 12.* Qualitative comparison of PointPatch + FF (upper row) and PointPatch (lower row) policies on three RoboCasa tasks. Policies trained without Fourier features have difficulty learning the demonstration data and carrying out complex movements with precision. Time proceeds from left to right in each row.

## A.6. Real World Results

Table 8 shows per-task scores for each real world task. In order to arrive at more fine-grained results, we report a task-dependent score rather than a simple success rate, where the maximum score for each task is given in the rightmost column of Table 8. Since policies are very sensitive to the initial scene configuration and the experiments are not blind, it is easy to introduce bias in the results. To minimize this, we employ an alternating testing scheme, where we alternate between rollouts of our method and the baseline. This allows the human operator to ensure a consistent scene configuration for both methods.

Table 9 shows additional results collected on the Cup-Stacking task with cups of different sizes. We note that these are results are distinct from those of Table 8, which is why the scores of each method differ between the two tables. However, the policy with Fourier features is still clearly better in both cases.

*Table 8.* Average scores across 16 rollouts on our four real-world experiments. Adding Fourier features results in significant improvements across all tasks, with the greatest gain seen in the Drawer task.

| Task | Demos Per Task | Methods with Scores | | |
|---|---|---|---|---|
| | | RGB + PointPatch (pretrained) | + FF | Max |
| Drawer | 102 | 0.3125 | **1.625** | 4 |
| Cup-Stacking | 80 | 0 | **0.625** | 2 |
| Arranging | 100 | 0.3125 | **1.3125** | 4 |
| Folding | 75 | 1.3125 | **1.6875** | 3 |

*Table 9.* Average scores across 10 additional rollouts on each pair of cups used in the Cup-Stacking task. The gap between the baseline and the policy with Fourier features widens as cup size decreases. For the smallest cups, neither policy is able to solve this challenging task and the advantage disappears.

| Cup Pair | Diameter of Smaller/ Bigger Cup (cm) | Methods with Scores (out of 2.0) | |
|---|---|---|---|
| | | RGB + PointPatch (pretrained) | + FF |
| Yellow/Orange | 7.5/8.0 | 1.1 | 1.4 |
| Blue/Purple | 6.0/7.0 | 0.5 | 1.6 |
| Red/Blue | 5.5/6.0 | 0.5 | 0.6 |
| Orange/Red | 5.0/5.5 | 0.5 | 0.4 |

## A.7. Spectral Analysis

To evaluate the spectral sensitivity of Fourier features, we consider the frequency response of different trained and untrained models with and without Fourier features. Specifically, we consider a random trajectory from the demonstrations for CoffeePressButton in RoboCasa. For each time step, we construct a $k$-nearest neighbor graph from the input point cloud, and then weight the resulting edges using the Zelnik-Manor product (Zelnik-Manor & Perona, 2004). We compute the eigenvalue decomposition of the Symmetric Normalized Laplacian, of which the eigenvalues typically lie in $\lambda \in [0, 2]$ (Chung, 1997). We utilize a sparse neighborhood of $k = 8$ to consider the local data manifold (Von Luxburg, 2007), ensuring the Laplacian remains sensitive to high-frequency geometric features. Given this graph and its Laplacian, we transform the input-output gradients onto the resulting graph Fourier basis and plot the model's response across the spectrum. Figure 8 visualizes the averaged and binned spectrum. We find that Fourier features increase the model's sensitivity to a wide range of frequencies, including high-frequency signals with $\lambda > 1.0$ that are otherwise suppressed by the low-pass bias of standard coordinate inputs.

To demonstrate the increase sensitivity to high frequency signals specifically, we consider a simple untrained PointNet consisting of a three-layer MLP followed by global attention pooling and a two-layer output MLP. The PointNet consumes 1024 randomly sampled points sampled from a unit ball, and we determine the gradient of the sum of all node's scalar output w.r.t. all input coordinates and average them to obtain a sensitivity of the output w.r.t. the graph input. We repeat this setup for 100 randomly sampled point clouds. This input-output saliency is then projected onto the graph Fourier basis of the symmetric normalized Laplacian using Zelnik-Manor local scaling (Zelnik-Manor & Perona, 2004). Through this decomposition of the saliency into the eigenvectors, we split the total gradient into directional derivatives corresponding to different wavelengths. Due to the use of the Symmetric Normalized Laplacian, the frequency domain is normalized to $(0, 2)$ where the normalized frequency 1 distinguishes between low and high frequencies. Plotting the eigenvalue strength of the saliency for a model with and without Fourier features in Figure 13, we can observe a decrease in sensitivity to low frequency changes and an increase to higher frequencies. Since the saliency map equates to the linear term of a first-order Taylor approximation of the model network, this confirms that adding Fourier features counteracts the spectral bias of the used MLPs.

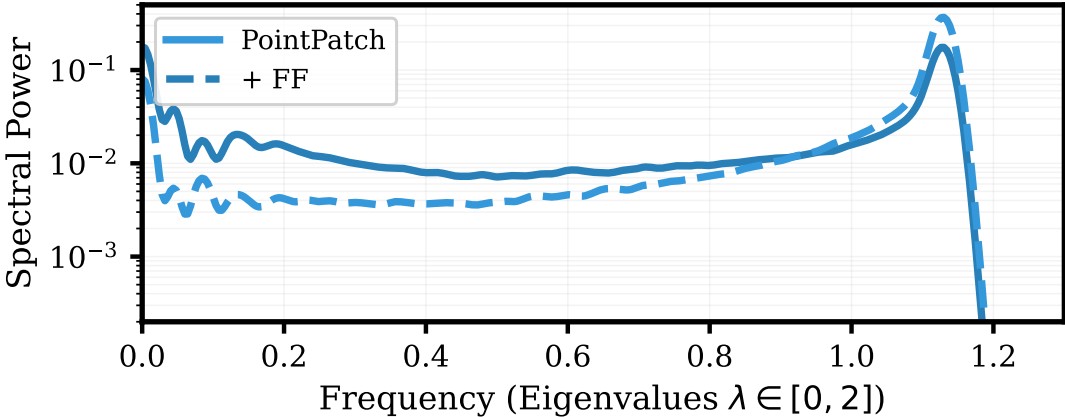

*Figure 13.* **Graph Spectral Sensitivity Comparison**. We consider a toy problem where we study the point-wise gradient of the sum of model outputs of an untrained PointNet on a point cloud of a sphere. By projecting these gradients onto the basis of a Symmetric Normalized Laplacian constructed with Zelnik-Manor local scaling, we observe that the vanilla architecture (dashed) is inherently biased toward low-frequency geometric components. In contrast, the addition of log-spaced Fourier features (solid line) amplifies the model's sensitivity to high-frequency manifold details ($\lambda > 1.0$), even before any task-specific training occurs.

