# OpenReview forum: "Fourier Features Let Agents Learn High Precision Policies with Imitation Learning"
_ICML.cc/2026/Conference — ICML 2026 regular_

### Official Review · Reviewer_t514 · 2026-02-16

**Soundness:** 3
**Presentation:** 3
**Significance:** 2
**Originality:** 2
**Overall Recommendation:** 4
**Confidence:** 3

**Summary:**

The contribution of this work, as I understand it, is an evaluation of the effect that fourier feature encodings have on the success rate of imitation learning robotic tasks, when they are added as encodings to point clouds. Further analysis was done to investigate the spectral bias of point cloud encoders with and without fourier feature inputs, to determine if greater attention is paid to high frequency features when using fourier encodings. Evaluation was done on a variety of relevant baselines in simulation, and on a single baseline in the real world.

**Compliance With Llm Reviewing Policy:**

Affirmed.

**Final Justification:**

With additional results and analysis from the rebuttal, I feel the contribution of this paper is better supported. I thank also the authors for giving more clarity around their contribution.

**Key Questions For Authors:**

1. Which tasks saw the largest improvements from fourier features, and were there any patterns to these tasks?
2. What is the main intended takeaway from Figure 7, and is there any way you could make it clearer?
3. Did you investigate further using full RGB for all baselines, rather than greyscale?

**Limitations:**

I would appreciate more discussion of the limitations of your study, albeit briefly.

**Strengths And Weaknesses:**

The main strengths of this work were the detailed evaluation, the spectral analysis, and the ablation. The evaluation, especially in simulation, was well presented and detailed. The selection of baselines was good, and the evaluation methods made sense. The real world evaluation was appreciated, and contained the most convincing results. The spectral analysis was particularly interesting, and important to the main argument of the paper. The ablation contains useful information to practitioners would might want to use fourier features in their own work. In general, I found the paper well written and easy to follow. I sincerely thank the authors for their submission, which I enjoyed reading.

The main limitation of this work is the contribution. Whilst the work is certainly interesting, well executed, and has value to the community, the contribution could be improved in several aspects. In particular, since the paper does not present a novel method, I felt that the rigour of the analysis was very important. Here are several recommendations to improve the paper:
* The contribution is broken up into three points in the main text, in summary: 1) adding fourier features (FF) to architectures and showing how this changes their spectral bias; 2) showing FF improve imitation learning success rates; 3) an ablation over key hyperparameters.
* For contribution point (1), I consider that 'adding FF to architectures' be dropped and that this point should refer mainly to how FF affect the spectral bias. This is because FF have already been presented across a variety of architectures in existing literature. Whereas, the spectral analysis is a particularly interesting argument of the paper, aiming to demonstrate the same improvements in the imitation learning domain which have already been demonstrated with NeRFs. However, I recommend that the spectral analysis be expanded and improved, that make this contribution point stronger. Currently, I found this analysis hard to follow. For Figure 7, I did not understand how the text corresponded to the figure. When you comment 'We find that Fourier features increase the sensitivity of the network to high frequencies, as well as low ones', what does this refer to specifically? Then, the statement, 'Since the sensitivites of all networks increase during training, Fourier features allow networks to learn the data more quickly.' I did not understand how this was supported by the results. Since this is such an important point for your contribution, I think it should be made much clearer and given more space in the text. The figure itself seems to suggest a lower sensitivity across all frequencies, and the text does not help the reader understand why this would be the case, or how this supports your argument. Perhaps using the normalisation across power would help show the reader better what is happening.
* Related to the previous point, the initialisation experiment in Appendix A.5 is interesting and well presented, and in its current form I felt it supported the claims made in the paper more than Figure 7. Since this is such an important point for your contribution, it could be worth adding this analysis into the main text.
* Related to the previous point, the tasks upon which fourier features had the greatest difference on performance during the simulated and real world experiments could be highlighted. Could you quantify the 'precision' required in some way? Or vary the size of the object in the interaction and illustrate that the fourier features improved performance for the smallest objects? In general, this would support your contribution.
* For contribution point (2), I think that the simulated experiments are well done, and the real world experiments are also very valuable. Overall, I consider this the strongest contribution. However, I would suggest that this contribution could be strengthened by more seeds and addition of statistical analysis. Were the addition of fourier features statistically significant? For how many of the benchmark tasks did they provide a statistically significant improvement? Since this paper positions as a study, I do feel that this is an essential element.
* The real world experiments were convincing, as well as the real world videos. Expanding on the real world experiments would add to the contribution by providing more detailed information about how fourier features improve performance. For example, examining the 'gap' between with and without fourier features in the real world as object size varied, or measuring the 'gap' across more baselines. Or, to support the qualitative analysis, you could characterise the actual smoothness of the actions and report it in the text.
* For contribution point (3), the ablation currently includes jitter, wavelengths, and fourier feature implementation. Statistical analysis would be greatly appreciated, to illustrate whether features result in a statistically significant improvement (or not, in the case of wavelengths). Since there are only 3 seeds, for jitter I would suggest that no conclusions can be made, even for reduced variance. Therefore, I think a more systematic and statistical ablation would improve the contribution.

---

> ### Author Rebuttal · Authors · 2026-03-31
>
> We kindly thank the reviewer for their feedback and kind words. The reviewer highlights the paper’s detailed evaluation, selection of baselines, and interesting analysis. We especially thank the reviewer for their numerous suggestions for improvement. We respond to each of these suggestions and requests for clarification below.
>
> 1. “FF have already been presented across a variety of architectures in existing literature.”
>
>     FFs for point clouds have only been presented in one prior work in the literature, so our addition of FFs to various common architectures is novel (please also see response to Reviewer ETiq, point #1). However, we will expand the spectral analysis in the final version.
>
> 2. “I found this analysis hard to follow. For Figure 7…”
>
>     Thank you for pointing out this confusing passage. Figure 7 shows that FFs increase the sensitivity of the network to all frequencies, given that all the spectra have roughly the same profile and are mainly just separated by a vertical offset. We can conclude that this is desirable, because all policies appear to become more sensitive to all frequencies during training (compare corresponding lines in Figure 7 before training (right) to after training (left)). Therefore we hypothesize that adding FFs gives the network a “head start” during training, because it does not need to spend as much “time” increasing its sensitivity. Normalizing the power sensitivity spectra would hide the orders of magnitude difference in their power, therefore we do not normalize.
>
> 3. “the initialisation experiment in Appendix A.5…”
>
>     Thank you for the positive feedback, we will incorporate it into the main paper.
>
> 4. “Could you quantify the 'precision' required in some way?”
>
>     We applied Gaussian random jitter with a fixed magnitude to the point clouds during training and rollout. We trained policies with sigma=0.002,0.005,0.01,0.02,0.05. The highest sigma values effectively removed any fine geometric information in the point cloud. The table below presents the relative success rates of policies with sigma=0.05 compared to the baseline (w/ FF). The final column shows the relative success of policies w/o FF for comparison.
>
>     |  | PointPatch w/ FF (% success) | +sigma=0.05 (relative) | -FF (relative) |
>     | --- | --- | --- | --- |
>     | CoffeeSetupMug | 0.7 | 0.00 | 0.00 |
>     | OpenSingleDoor | 15.6 | 0.24 | 0.06 |
>     | CoffeePressButton | 37.6 | 0.34 | 0.41 |
>     | OpenDrawer | 11.4 | 0.34 | 0.06 |
>     | OpenDoubleDoor | 19.0 | 0.42 | 0.00 |
>     | TurnOnSinkFaucet | 35.7 | 0.53 | 0.43 |
>     | CloseSingleDoor | 53.6 | 0.56 | 0.43 |
>     | TurnOffMicrowave | 40.8 | 0.59 | 0.39 |
>     | TurnOnStove | 39.2 | 0.69 | 0.31 |
>     | overall | 33.5 | 0.70 | 0.38 |
>     | TurnOffStove | 20.6 | 0.72 | 0.39 |
>     | TurnOffSinkFaucet | 63.6 | 0.77 | 0.44 |
>     | TurnSinkSpout | 63.4 | 0.80 | 0.62 |
>     | CloseDoubleDoor | 31.5 | 0.85 | 0.06 |
>     | CloseDrawer | 70.3 | 0.88 | 0.48 |
>     | TurnOnMicrowave | 31.2 | 0.93 | 0.25 |
>     | CoffeeServeMug | 1.2 | 7.97 | 0.52 |
>
>     Overall, there seems to be only moderate correlation between the effect of removing fine geometry from the observation and of removing FFs from the policy. It appears difficult to conclusively predict the benefit of adding FFs purely from the task characteristics. However, this is consistent with the spectral analysis discussed above, since we show that FFs also increase the sensitivity to low frequencies, which at least partly explains in the increase in performance even without any fine geometry to extract.
>
> 5. “more seeds and addition of statistical analysis”
>
>     We now apply bootstrapping[1] by sampling from 3 seeds with replacement and sampling 50 (100 for ManiSkill) episodes from each checkpoint 10k times. Since we observe considerable variance in the evaluation results due to the binary success metric, we have rerun all evaluations with 50 episodes instead of 20 (100 instead of 50 for ManiSkill). We now report interquartile mean and 95% confidence intervals. We will post the updated results in a follow-up comment as soon as we have them. Our updated results so far indicate that the relative performance gain across task suites is remarkably stable. Furthermore, we are happy to increase the number of seeds from 3 to 5 for the camera ready version.
>
>     [1] Agarwal et al. "Deep reinforcement learning at the edge of the statistical precipice." NeurIPS 2021.
>
> 6. “Expanding on the real world experiments”
>
>     We really appreciate the excellent suggestions to improve our experiments. We are currently working on additional real world experiments with different sized objects which we hope to report on before the end of the discussion period.
>
> 7. “a more systematic and statistical ablation”
>
>     We agree. As mentioned above, we will increase the number of seeds everywhere from 3 to 5, and we will reevaluate the claim of reduced variance for VariableJitter.

---

> > ### Author Rebuttal · Reviewer_t514 · 2026-04-03
> >
> > I thank the authors for their detailed responses. My concerns are addressed and I look forward to seeing some update results. I sincerely believe the additional analysis can strengthen the paper and I will improve my score.

---

> > > ### Author Response · Authors · 2026-04-07
> > >
> > > We thank the reviewer once again for their valuable feedback, which helps us improve our manuscript.
> > >
> > > → 5. We have repeated the evaluation of all methods and ablations (including the new PointTransformer implementation, see comments by Reviewer iFRk) now with 50 episodes for RoboCasa and 100 for ManiSkill3. We compute bootstrap confidence intervals by sampling 10000 experiments, each consisting of sampling 3 seeds with replacement and 50 (respectively 100) episodes from each checkpoint of those seeds. We take the maximum success rate across checkpoints and average across seeds, then average across all tasks. We report the interquartile mean (IQM) and 95% confidence bounds. The final version of the paper will include 5 seeds for all results.
> > >
> > > 1. Main results (Figure 4). FFs provide consistent and significant benefits for RoboCasa across architectures. With PointPatch for example, we note statistically significant improvement (non-overlapping confidence bounds) in 14 of 16 tasks. For ManiSkill3, FFs provide a slight but not statistically significant benefit, indicating performance saturation.
> > >
> > >
> > >     |  | RoboCasa |  |  | ManiSkill3 |  |  |
> > >     | --- | --- | --- | --- | --- | --- | --- |
> > >     |  | baseline (% success) | + FF (% success) | # improved (of 16) | baseline (% success) | + FF (% success) | # improved (of 4) |
> > >     | PointPatch | 12.7 ± 1.4 | 33.5 ± 1.9 | 14 | 47.3 ± 3.2 | 55.3 ± 2.9 | 2 |
> > >     | PointPatch-attn | 16.1 ± 1.5 | 29.2 ± 1.7 | 8 | 50.7 ± 2.4 | 54.5 ± 3.3 | 0 |
> > >     | PCM | 19.1 ± 1.5 | 30.0 ± 1.8 | 6 | 62.8 ± 2.4 | 65.5 ± 3.2 | 0 |
> > >     | DP3 | 14.7 ± 1.6 | 28.2 ± 1.8 | 10 | 62.8 ± 2.5 | 63.0 ± 2.5 | 0 |
> > >     | PointTransformer | 23.4 ± 1.6 | 28.8 ± 1.9 | 2 | 47.7 ± 3.6 | 50.4 ± 3.1 | 0 |
> > >     | PointPatch+RGB | 21.5 ± 1.9 | 26.5 ± 2.3 | 1 | 63.8 ± 2.4 | 65.2 ± 2.4 | 0 |
> > >     | PointPatch+RGB (pretrained) | 44.8 ± 1.9 | 47.4 ± 2.5 | 0 | 69.5 ± 2.4 | 68.8 ± 2.2 | 0 |
> > >     | PointMap | 39.9 ± 2.1 | - | - | 62.2 ± 2.7 | - | - |
> > >     | RGB | 16.4 ± 1.6 | - | - | 61.6 ± 3.0 | - | - |
> > > 2. Parameter study over frequency configuration (Figure 6). FFs are robust to different minimum wavelengths and numbers of frequency bands. The only effect that approaches statistical significance is a reduction in performance with L=8 wavelengths.
> > >
> > >
> > >     | method | success rate (%) |
> > >     | --- | --- |
> > >     | ours (L=16, λ_min=0.02) | 41.5 ± 3.1 |
> > >     | L=8 | 36.9 ± 2.6 |
> > >     | L=16 | 41.5 ± 3.1 |
> > >     | L=32 | 40.0 ± 2.9 |
> > >     | λ_min=0.01 | 40.7 ± 3.3 |
> > >     | λ_min=0.02 | 41.5 ± 3.1 |
> > >     | λ_min=0.05 | 41.1 ± 2.6 |
> > >     | λ_min=0.1 | 40.9 ± 2.9 |
> > >     | λ_min=0.2 | 39.7 ± 2.7 |
> > > 3. Parameter study over jitter and RFFs (Table 1). We thank the reviewer for prompting us to reassess this parameter study, now finding no significant difference between jitter variants.
> > >
> > >
> > >     | method | success rate (%) |
> > >     | --- | --- |
> > >     | ours | 41.5 ± 3.1 |
> > >     |  |  |
> > >     | no FFs, no jitter | 17.7 ± 2.2 |
> > >     | no FFs, random jitter | 16.3 ± 2.0 |
> > >     | no FFs, VariableJitter | 17.7 ± 2.4 |
> > >     | FFs, no jitter | 40.6 ± 3.0 |
> > >     | FFs, random jitter | 38.0 ± 2.7 |
> > >     |  |  |
> > >     | log-spaced + SPE | 37.1 ± 2.6 |
> > >     | RFF | 23.9 ± 2.5 |
> > >     | RFF + learned | 23.6 ± 2.3 |
> > >     | RFF + SPE | 22.1 ± 2.4 |
> > >     | RFF + Cartesian | 23.1 ± 2.4 |
> > >
> > > → 6. We collected 10 additional episodes on each pair of cups used for the Cup Stacking task and report the results according to cup size. In order to keep the comparison fair while varying starting conditions, we alternate rollouts of the policy and baseline, ensuring that each initial condition is repeated for both. Since in our earlier experiments the baseline policy achieved a score of 0 on Cup Stacking (see Table 6), we now use less challenging starting conditions, resulting in improvements for both policies. While results with the largest cups (7.5/8 cm) are comparable, FFs maintain good performance for the Blue/Purple cups (6/7 cm), where the baseline fails. This demonstrates how FFs can increase the policy’s sensitivity to fine geometric details. Under ~6cm, both policies suffer greatly, potentially limited by depth sensor accuracy and calibration error.
> > >
> > > | Subtask | Diameter of Smaller/Bigger Cup (cm)      | baseline (score out of 2.0)      | + FF (score out of 2.0)      |
> > > | --- | --- | --- | --- |
> > > | Yellow/Orange | 7.5/8.0 | 1.1 | 1.4 |
> > > | Blue/Purple | 6.0/7.0 | 0.5 | 1.6 |
> > > | Red/Blue | 5.5/6.0 | 0.5 | 0.6 |
> > > | Orange/Red | 5.0/5.5 | 0.5 | 0.4 |
> > >
> > > If there are any remaining questions or concerns, please let us know and we will try to respond before the end of the discussion phase.

---

### Official Review · Reviewer_ETiq · 2026-03-09

**Soundness:** 3
**Presentation:** 3
**Significance:** 3
**Originality:** 2
**Overall Recommendation:** 4
**Confidence:** 4

**Summary:**

This paper tackles the problem of spectral bias in imitation learning from point clouds. The authors propose to feed point coordinates through a NeRF-style Fourier feature mapping before passing them to a point cloud encoder. They argue that this helps the network capture high-frequency geometric details, which is crucial for fine manipulation tasks. The method is evaluated on several benchmarks (RoboCasa, ManiSkill3, and real-world tasks) using multiple point cloud architectures (PointPatch, DP3, PCM). They also incorporate a recent data augmentation technique called VariableJitter and analyze frequency sensitivity via Graph Fourier Transform.

**Compliance With Llm Reviewing Policy:**

Affirmed.

**Final Justification:**

The author has addressed my concerns and provided supporting evidence. so I would keep my positive score.

**Key Questions For Authors:**

1. Table 1 shows something interesting: adding VariableJitter doesn't change the average performance (both around 0.38), but it drops the standard deviation from 0.030 to 0.006. I'm curious why this happens.

2.Denser point clouds contain richer geometric detail, so Fourier features should matter more. In other words, if I downsample to 1024 points, the gain from Fourier features should shrink or disappear. Have you tested this?

3.The paper argues that fine manipulation tasks require high-frequency decision boundaries, and that Fourier features help networks learn them. I'm not entirely convinced that we have direct evidence these tasks actually need high-frequency decisions. Fine manipulation → must need high frequencies → Fourier features help → therefore the tasks needed high frequencies. This feels a bit like circular reasoning.

**Limitations:**

The authors briefly mention limitations in Section 9 ("Impact Statement"), but this section is more about societal impact than technical limitations.

**Strengths And Weaknesses:**

Soundness:
The paper is technically sound. The experiments are well-designed and thorough, covering multiple simulation benchmarks (RoboCasa, ManiSkill3), real-world tasks, and several point cloud architectures (PointPatch, DP3, PCM).

Presentation:
The paper is clearly written and well-structured.

Significance:
Enabling imitation learning policies to capture high-frequency decision boundaries needed for fine manipulation

Originality:
This is where the paper is weakest. All core components are taken directly from prior work: Fourier features (Mildenhall et al., Tancik et al.), VariableJitter (Gyenes et al., 2025), and the GFT analysis methodology (Miao et al., 2024). The paper's contribution is essentially a validation that these existing techniques work in the new context of point-cloud-based imitation learning. Spectral bias itself is a well-known property of neural networks (Rahaman et al., 2019), so finding that it affects IL policies is not a surprising discovery. The paper does not offer new insights beyond confirming what one might expect given the literature. The claimed architecture-agnostic advantages stem directly from existing techniques, not from any novel contribution in this work.

---

> ### Author Rebuttal · Authors · 2026-03-31
>
> We thank the reviewer for their detailed and critical review, and the chance to improve our manuscript by addressing their concerns. The reviewer notes that the paper is clearly written, well-structured, and technically sound, with well-designed and thorough experiments. We respond to the reviewer’s criticisms and questions below.
>
> 1. “finding that [spectral bias] affects IL policies is not a surprising discovery. The paper does not offer new insights”
>
>     While it may seem intuitively obvious that adding FFs to improve sensitivity to high frequencies would also improve point cloud-based imitation learning, it is essentially unseen in the literature. Even recent foundation models for point cloud representations[1-3] do not use FFs. We are only aware of a single work[4] in 3D imitation learning that mentions FFs, and no works using FFs for point cloud segmentation, classification, or generation. In contrast, we present a systematic study of FFs in many common IL architectures.  (While several works apply Fourier *transforms*, this is fundamentally different from an FF projection, as it does not resolve the spectral bias.)
>
>     Furthermore, we argue that our paper presents some surprising insights. One might intuitively expect FFs to be susceptible to overfitting. Although the point clouds in real world experiments have plenty of noise and spurious artifacts, the policies show no evidence of overfitting on these features, and are qualitatively much smoother. While other works have found FFs to be sensitive to hyperparameters in domains like image generation[5], we observe the opposite. By showing strong general performance for IL, we provide other researchers the basis for adopting FFs in their architectures.
>
>     [1] Chen et al. "Sugar: Pre-training 3d visual representations for robotics." CVPR 2024.
>
>     [2] Li et al. "Pointvla: Injecting the 3d world into vision-language-action models." RAL 2026.
>
>     [3] Jia et al. "Lift3d policy: Lifting 2d foundation models for robust 3d robotic manipulation." CVPR 2025.
>
>     [4] Wilcox et al. "Adapt3R: Adaptive 3D Scene Representation for Domain Transfer in Imitation Learning." CoRL 2025.
>
>     [5] Song et al. "Improved Techniques for Training Consistency Models." ICLR 2023.
>
> 2. “adding VariableJitter doesn't change the average […], but it drops the standard deviation”
>
>     We hypothesize that this data augmentation stabilizes the training process. However, only 3 random seeds were tested, so it may be a statistical outlier. As noted in the response to Reviewer t514, we will add more seeds and increase the number of evaluation rollouts for the final version.
>
> 3. “Denser point clouds contain richer geometric detail, so FFs should matter more”
>
>     To reduce point cloud sizes while retaining their overall structure, we increase the voxel size used for voxel downsampling. Please see the table below for success rates with and without FFs for different voxel sizes, as well as the approximate point clouds sizes resulting from each voxel size.
>
>     | voxel size (cm) | PointPatch w/ FF (% success) | PointPatch (% success) | Approx. Point Count |
>     | --- | --- | --- | --- |
>     | 1.0 | 39.6 ± 2.6 | 17.0 ± 2.2 | 18k |
>     | 1.5 | 38.9 ± 2.7 | 18.5 ± 2.0 | 15k |
>     | 2.0 | 36.2 ± 2.5 | 19.4 ± 2.2 | 9k |
>     | 3.0 | 32.9 ± 2.4 | 16.1 ± 2.2 | 4.5k |
>     | 5.0 | 26.0 ± 2.5 | 19.0 ± 2.1 | 2k |
>
>     As hypothesized by the reviewer, the benefit of FFs narrows for smaller point clouds, while policies without FFs seem largely unaffected. This may be because increasing voxel size reduces the fine geometric details that the FF policies condition on. We find it difficult to downsample a point cloud so aggressively while ensuring that essential geometric information is still intact. We thank the reviewer for suggesting this experiment and are happy to add it to the revised paper to provide a more complete picture of the use cases and limitations of FFs.
>
> 4. “I'm not entirely convinced that […] these tasks actually need high-frequency decisions. Fine manipulation → must need high frequencies → FFs help → therefore the tasks needed high frequencies. This feels a bit like circular reasoning”
>
>     We apologize for any confusion. Our motivating hypothesis was that IL policies may benefit from high frequency decision boundaries, especially if tasks involve fine manipulation. Our results demonstrate that FFs improve success rates, and our spectral analysis shows that they increase the network’s sensitivity *across the frequency spectrum*. Our main finding is that FFs improve policy performance consistently across a broad range of tasks, architectures, and hyperparameters. Some tasks appear to benefit from increased sensitivity to higher frequencies, some appear to benefit from an overall increase in sensitivity. Please see our response to Reviewer t514, point #4 for a thorough study of this. We will revise this to make it clearer.

---

> > ### Author Rebuttal · Reviewer_ETiq · 2026-04-01
> >
> > The author has addressed my concerns and provided supporting evidence, so I will improve my score.

---

> > > ### Author Response · Authors · 2026-04-07
> > >
> > > We thank the reviewer again for their comments and criticisms, as we sincerely believe they have made the manuscript better. We will revise the sections of the text highlighted by the reviewer and add the parameter study on point cloud size to the final paper. Lastly, we would like to extend our best wishes to the reviewer!

---

### Official Review · Reviewer_iFRk · 2026-03-13

**Soundness:** 3
**Presentation:** 3
**Significance:** 3
**Originality:** 3
**Overall Recommendation:** 4
**Confidence:** 4

**Summary:**

This paper argues that a key shortcoming in existing 3D-based imitation learning frameworks is the spectral bias of the architectures, which makes the models hard to learn accurate geometry for high-precision tasks. To address this, the paper incorporates Fourier feature mappings into pointcloud encoders and shows that this resolves the spectral bias of various encoder networks. It also shows that this improves the performances of imitation learning in simulations and real-robot environments. Ablations also show that the Fourier features don't need any additional regularizations and is robust to choice of hyperparameters.

**Compliance With Llm Reviewing Policy:**

Affirmed.

**Final Justification:**

My concerns are well resolved after seeing the authors' additional experiments and explanations, so I would keep my positive score.

**Key Questions For Authors:**

- Perhaps it'd be good to have more explanations of how the tasks are chosen from Robocasa.
- Can the proposed Fourier feature mapping also be incorporated into point transformers (Zhao et. al., 2021)?

**Limitations:**

Yes, broader impacts are discussed in the paper.

**Strengths And Weaknesses:**

Strengths:
- This is an interesting paper. It suggests a different perspective than many exsiting works: most works in policy learning focus on the design of the action prediction network branches, but this paper instead addresses the perspective of scene encoding -- this is indeed an important perspective that can be easily neglected.
- The proposed Fourier feature mapping is simple and versitle that can be easily incorporated into different pointcloud encoders.
- I like the detail that the wavelengths are consistent instead of handcrafted for each senario, and the network stability is encouraged through data augmentation.

Weaknesses:
- Overall, I tend to feel that the experimental results are supportive of the conclusiIons. But I am a bit concerned of the Maniskill results, where the improvements are marginal (and for PCM, the Fourier feature result is slightly lower than the baseline). On one hand, I may agree that the Maniskill tasks are more general and less high-precision, and thus they may not be the best showcase for the proposed method; but on the other hand, the Robocasa tasks are manually selected from the full Robocasa task sets, this may make the results biased.

---

> ### Author Rebuttal · Authors · 2026-03-31
>
> We would like to thank the reviewer for their kind review and positive feedback. We appreciate that they highlighted the simplicity and versatility of the method, and our use of consistent hyperparameters across all experiments. Below, we address the weaknesses and questions posed by the reviewer.
>
> 1. “I am a bit concerned of the Maniskill results, where the improvements are marginal”
>
>     As pointed out by the reviewer, the ManiSkill tasks do not require as much precision, and we find evidence of performance saturation even without FFs. However, we note that there are extremely few instances where adding FFs results in significantly worse performance (e.g. for ManiSkill, only PCM on PokeCube).
>
> 2. “Perhaps it'd be good to have more explanations of how the tasks are chosen from Robocasa.”
>
>     We regret that our choice of RoboCasa tasks gave the appearance of some sort of task tuning, as this was not the case. In fact, we only exclude two groups of tasks. First, we drop the NavigateKitchen task because the mobile robot base results in an incompatible action space with the other tasks. Second, we drop the 8 pick and place tasks simply because early experiments showed negligible success rates both with and without FFs. Perhaps this could have been improved by tuning hyperparameters or training with fewer tasks, but we opted to introduce a second task suite (ManiSkill) instead. All the remaining 16 atomic tasks are presented in our results. We will make the task selection process more transparent in the final version of the paper.
>
> 3. “Can the proposed FF mapping also be incorporated into point transformers (Zhao et. al., 2021)?”
>
>     We adapt the implementation given in the examples for [torch_geometric](https://github.com/pyg-team/pytorch_geometric/blob/master/examples/point_transformer_classification.py). Since the node position inputs are used for Farthest Point Sampling and KNN queries, we do not modify this input in any way. Instead, we also pass the point positions as the node features, optionally with a FF projection. Because of the high compute costs of this architecture, we train each policy on 2-3 tasks instead of the typical 8.
>
>     We refer to the results in the table below (please note that we now report interquartile means and bootstrap confidence intervals, see response to Reviewer t514 for full description):
>
>     | RoboCasa | PointTransformer +FF (%) | PointTransformer (%) |
>     | --- | --- | --- |
>     | CoffeePressButton | 27.0 ± 7.7 | 24.4 ± 7.0 |
>     | TurnOnMicrowave | 34.2 ± 7.8 | 20.1 ± 6.6 |
>     | TurnOffMicrowave | 39.7 ± 7.0 | 24.6 ± 8.7 |
>     | TurnOnSinkFaucet | 40.7 ± 8.6 | 25.9 ± 6.7 |
>     | TurnOffSinkFaucet | 54.7 ± 12.0 | 53.6 ± 9.7 |
>     | TurnSinkSpout | 59.3 ± 8.7 | 52.6 ± 8.7 |
>     | TurnOnStove | 38.3 ± 7.1 | 31.3 ± 8.1 |
>     | TurnOffStove | 21.6 ± 6.4 | 17.8 ± 5.5 |
>     | overall | 39.5 ± 2.9 | 31.4 ± 2.7 |
>
>     | ManiSkill | PointTransformer +FF (%) | PointTransformer (%) |
>     | --- | --- | --- |
>     | PokeCube-v1 | 56.1 ± 8.4 | 57.8 ± 7.2 |
>     | PushCube-v1 | 72.5 ± 5.5 | 72.2 ± 8.8 |
>     | PullCube-v1 | 61.3 ± 6.0 | 51.6 ± 7.6 |
>     | RollBall-v1 | 11.5 ± 2.8 | 9.7 ± 4.0 |
>     | overall | 50.3 ± 3.0 | 47.7 ± 3.6 |
>
>     Adding FFs results in a statistically significant increase for TurnOnMicrowave, TurnOffMicrowave, and TurnOnSinkFaucet, and a slight benefit on the other tasks. On no task does it result in worse performance. Overall, it produces a signficant benefit on the RoboCasa on average and a slight benefit on the ManiSkill tasks. These results underscore that FFs are a versatile technique that can benefit essentially any point cloud architecture. We will continue these experiments and add results on the remaining RoboCasa tasks to the final paper.
>
>
> We thank the reviewer again for taking the time to read our manuscript and make suggestions for improvement. We look forward to engaging in productive discussion during the upcoming week.

---

> > ### Author Rebuttal · Reviewer_iFRk · 2026-04-05
> >
> > Thank the authors for their explanations and additional experiments! My concerns are well resolved.

---

> > > ### Author Response · Authors · 2026-04-05
> > >
> > > We thank the reviewer again for their helpful feedback and suggestions. If there is anything else we can do to improve the manuscript and potentially earn a higher score, please do not hesitate to let us know. Otherwise, we extend our best wishes!

---

### Decision · Program_Chairs · 2026-04-30

**Decision:**

Accept (regular)

**Comment:**

All three reviewers agreed that the paper provides a valuable contribution by demonstrating that Fourier feature projections consistently improve point cloud-based imitation learning across diverse architectures and benchmarks. The main concerns raised were about limited originality, marginal improvements on ManiSkill tasks, and the need for stronger statistical analysis. The authors addressed these concerns thoroughly in their rebuttal. All three reviewers acknowledged their concerns were resolved and maintained or improved their positive scores.